# Quantifying the nitrogen isotope effects during photochemical equilibrium between NO and NO$_2$: implications for $\delta^{15}$N in tropospheric reactive nitrogen

Jianghanyang Li[1], Xuan Zhang[2], John Orlando[2], Geoffrey Tyndall[2] and Greg Michalski[1,3]

[1.] Department of Earth, Atmospheric and Planetary Sciences, Purdue University, West Lafayette, IN, 47907

[2.] Atmospheric Chemistry Observations and Modeling Lab, National Center for Atmospheric Research, Boulder, CO, 80301

[3.] Department of Chemistry, Purdue University, West Lafayette, IN, 47907

*Correspondence to*: Jianghanyang Li (li2502@purdue.edu)

**Abstract.** Nitrogen isotope fractionations between nitrogen oxides (NO and NO$_2$) play a significant role in determining the nitrogen isotopic compositions ($\delta^{15}$N) of atmospheric reactive nitrogen. Both the equilibrium isotopic exchange between NO and NO$_2$ molecules and the isotope effects occurring during the NO$_x$ photochemical cycle are important, but both are not well constrained. The nighttime and daytime isotopic fractionations between NO and NO$_2$ in an atmospheric simulation chamber at atmospherically relevant NO$_x$ levels were measured. Then, the impact of NO$_x$ level and NO$_2$ photolysis rate to the combined isotopic fractionation (equilibrium isotopic exchange and photochemical cycle) between NO and NO$_2$ were calculated. It was found that the isotope effects occurring during the NO$_x$ photochemical cycle can be described using a single fractionation factor, designated the Leighton Cycle Isotope Effect (LCIE). The results showed that at room temperature, the fractionation factor of nitrogen isotopic exchange is 1.0289±0.0019, and the fractionation factor of LCIE (when O$_3$ solely controls the oxidation from NO to NO$_2$) is 0.990±0.005. The measured LCIE factor showed good agreement with previous field measurements, suggesting that it could be applied in ambient environment, although future work is needed to assess the isotopic fractionation factors of NO + RO$_2$/HO$_2$ $\rightarrow$NO$_2$. The results were used to model the NO-NO$_2$ isotopic fractionations under several NO$_x$ conditions. The model suggested that isotopic exchange was the dominate factor when NO$_x$ >20 nmol mol$^{-1}$, while LCIE was more important at low NO$_x$ concentrations (<1 nmol mol$^{-1}$) and high rates of NO$_2$ photolysis. These findings provided a useful tool to quantify the isotopic fractionations between tropospheric NO and NO$_2$, which can be applied in future field observations and atmospheric chemistry models.

## 1. Introduction

The nitrogen isotopic composition ($\delta^{15}$N) of reactive nitrogen compounds in the atmosphere is an important tool in understanding the sources and chemistry of atmospheric $NO_x$ ($NO+NO_2$). It has been suggested that the $\delta^{15}$N value of atmospheric nitrate ($HNO_3$, nitrate aerosols and nitrate ions in the precipitation and snow) imprints the $\delta^{15}$N value of $NO_x$ sources (Elliott et al., 2009; Kendall et al., 2007) thus many studies have used the $\delta^{15}$N values of atmospheric nitrate to investigate $NO_x$ sources (Chang et al., 2018; Felix et al., 2012; Felix & Elliott, 2014; Gobel et al., 2013; Hastings et al., 2004, 2009; Morin et al., 2009; Park et al., 2018; Walters et al., 2015, 2018). However, there remain questions about how isotopic fractionations that may occur during photochemical cycling of $NO_x$ could alter the $\delta^{15}$N values as it partitions into $NO_y$ ($NO_y$ = atmospheric nitrate, $NO_3$, $N_2O_5$, HONO, etc., Chang et al., 2018; Freyer, 1991; Hastings et al., 2004; Jarvis et al., 2008; Michalski et al., 2005; Morin et al., 2009; Zong et al., 2017). Similarily, other complex reactive nitrogen chemistry, such as nitrate photolysis and re-deposition in ice and snow (Frey et al., 2009), may impact the $\delta^{15}$N of $NO_y$ and atmospheric nitrate. The fractionation between NO and $NO_2$ via isotope exchange has been suggested to be the dominant factor in determining the $\delta^{15}$N of $NO_2$ and ultimately atmospheric nitrate (Freyer, 1991; Freyer et al., 1993; Savarino et al., 2013; Walters et al., 2016). However, isotopic fractionations occur in most, if not all, $NO_x$ and $NO_y$ reactions, while most of these are still unknown or, if calculated (Walters and Michalski, 2015), unverified by experiments. Since the atmospheric chemistry of $NO_y$ varies significantly in different environments (e.g., polluted vs. pristine, night vs. day), the isotopic fractionations associated with $NO_y$ chemistry are also likely to vary in different environments. These unknowns could potentially bias conclusions about $NO_x$ source apportionment reached when using nitrogen isotopes. Therefore, understanding the isotopic

fractionations between NO and $NO_2$ during photochemical cycling could improve our
understanding of the relative role of sources versus chemistry for controlling the $\delta^{15}N$ variations
of atmospheric $NO_2$ and nitrate.
In general, there are three types of isotopic fractionation effects associated with $NO_x$
chemistry (Fig. 1A). The first type is the equilibrium isotopic effect (EIE), i.e., isotope exchange
between two compounds without forming new molecules (Urey, 1947, Bigeleisen and Mayer,
1947), which for nitrogen isotopes in the $NO_x$ system is the $^{15}NO + {}^{14}NO_2 \leftrightarrow {}^{14}NO + {}^{15}NO_2$
exchange reaction (Begun and Melton, 1956, Walters et al., 2016). The second type is the kinetic
isotopic effect (KIE) associated with difference in isotopologue rate coefficients during
unidirectional reactions (Bigeleisen & Wolfsberg, 1957). In the $NO_x$ system this KIE would
manifest in the oxidation of NO into $NO_2$ by $O_3/HO_2/RO_2$. The third type is the photochemical
isotope fractionation effect (PHIFE, Miller & Yung, 2000), which for $NO_x$ is the isotopic
fractionation associated with $NO_2$ photolysis. All three fractionations could impact the $\delta^{15}N$ value
of $NO_2$, and consequently atmospheric nitrate, but the relative importance of each may vary.
The limited number of studies on the EIE in the $NO_x$ cycle have significant uncertainties.
Discrepancies in the EIE for $^{15}NO + {}^{14}NO_2 \leftrightarrow {}^{14}NO + {}^{15}NO_2$ have been noted in several studies.
Theoretical calculations predicted isotope fractionation factors ($\alpha$) ranging from 1.035 to 1.042 at
room temperature (Begun & Fletcher, 1960; Monse et al., 1969; Walters & Michalski, 2015) due
to the different approximations used to calculate harmonic frequencies in each study. Likewise,
two separate experiments measured different room temperature fractionation factors of
1.028±0.002 (Begun & Melton, 1956) and 1.0356±0.0015 (Walters et al., 2016). A concern in both
experiments is that they were conducted in small chambers with high $NO_x$ concentrations
(hundreds of $\mu mol\ mol^{-1}$), significantly higher than typical ambient atmospheric $NO_x$ levels
(usually less than 0.1 µmol mol$^{-1}$). Whether the isotopic fractionation factors determined by these
experiments are applicable in the ambient environment is uncertain because of possible wall effects
and formation of higher oxides, notably $N_2O_4$ and $N_2O_3$ at these high $NO_x$ concentrations.

Even less research has examined the KIE and PHIFE occurring during $NO_x$ cycling. The

KIE of $NO + O_3$ has been theoretically calculated (Walters and Michalski, 2016) but has not been
experimentally verified. The $NO_2$ PHIFE has not been experimentally determined or theoretically
calculated. As a result, field observation studies often overlook the effects of PHIFE and KIE.
Freyer et al. (1993) measured $NO_x$ concentrations and the $\delta^{15}N$ values of $NO_2$ over a 1-year period
at Julich, Germany and inferred a combined $NO_x$ isotope fractionation factor (EIE+KIE+PHIFE)
of 1.018±0.001. Freyer et al. (1993) suggested that the $NO_x$ photochemical cycle (KIE and PHIFE)
tends to diminish the equilibrium isotopic fractionation (EIE) between NO and $NO_2$. Even if this
approach were valid, applying this single fractionation factor elsewhere, where $NO_x$, $O_3$
concentrations and actinic fluxes are different, would be tenuous given that these factors may
influence the relative importance of EIE, KIE and PHIFE (Hastings et al., 2004; Walters et al.,
2016). Therefore, to quantify the overall isotopic fractionations between NO and $NO_2$ at various
tropospheric conditions, it is crucial to know 1) isotopic fractionation factors of EIE, KIE and
PHIFE individually and 2) the relative importance of each factor under various conditions.

In this work, we aim to quantify the nitrogen isotope fractionation factors between NO and

$NO_2$ at photochemical equilibrium. First, we measure the N isotope fractionations between NO
and $NO_2$ in an atmospheric simulation chamber at atmospherically relevant $NO_x$ levels. Then, we
provide mathematical solutions to assess the impact of $NO_x$ level and $NO_2$ photolysis rate ($j(NO_2)$)
to the relative importance of EIE, KIE and PHIFE. Subsequently we use the solutions and chamber
measurements to calculate the isotopic fractionation factors of EIE, KIE and PHIFE. Lastly, using

102 the calculated fractionation factors and the equations, we model the NO-NO$_2$ isotopic

103 fractionations at several sites to illustrate the behavior of $\delta^{15}N$ values of NO$_x$ in the ambient

104 environment.


106 **2. Methods**

107  The experiments were conducted using a 10 m$^3$ Atmospheric Simulation Chamber at the

108 National Center for Atmospheric Research (see descriptions in Appendix A and Zhang et al.

109 (2018)). A set of mass flow controllers was used to inject NO and O$_3$ into the chamber. NO was

110 injected at 1 L min$^{-1}$ from an in-house NO/N$_2$ cylinder (133.16 μmol mol$^{-1}$ NO in ultra-pure N$_2$),

111 and O$_3$ was generated by flowing 5 L min$^{-1}$ zero-air through a flow tube equipped with a UV Pen-

112 Ray lamp (UVP LLC., CA) into the chamber. NO and NO$_2$ concentrations were monitored in real

113 time by chemiluminescence with a detection limit of 0.5 nmol mol$^{-1}$ (model CLD 88Y, Eco Physics,

114 MI) as were O$_3$ concentrations using an UV absorption spectroscopy with a detection limit of 0.5

115 nmol mol$^{-1}$ (model 49, Thermo Scientific, CO). In each experiment, the actual amounts of NO and

116 O$_3$ injected were calculated using measured NO$_x$ and O$_3$ concentrations after steady state was

117 reached (usually within 1 h). The wall loss rate of NO$_2$ was tested by monitoring O$_3$ (29 nmol mol$^{-1}$

118 ) and NO$_x$ (62 nmol mol$^{-1}$) over a 4-hour period. After the NO and NO$_2$ concentrations reached

119 steady state, no decrease in NO$_2$ concentrations was observed showing that chamber wall loss was

120 negligible.

121  Three experiments were conducted to measure the $\delta^{15}N$ value of the tank NO (i.e., the $\delta^{15}N$

122 value of total NO$_x$). In each of these experiments, a certain amount of O$_3$ was first injected into the

123 chamber, then approximately the same amount of NO was injected into the chamber to ensure 100%

124 of the NO$_x$ was in the form of NO$_2$ with little O$_3$ (<15 nmol mol$^{-1}$) remaining in the chamber, such

that the $O_3+NO_2$ reaction was negligible. The $NO_2$ in the chamber was then collected and its $\delta^{15}N$
value measured, which equates to the $\delta^{15}N$ value of the tank NO.

Two sets of experiments were conducted to separately investigate the EIE, KIE and PHIFE.

The first set of experiments was conducted in the dark. In each of these dark experiments, a range
of NO and $O_3$ ([$O_3$]<[NO]) was injected into the chamber to produce $NO-NO_2$ mixtures with
[NO]/[$NO_2$] ratios ranging from 0.43 to 1.17. The N isotopes of these mixtures were used to
investigate the EIE between NO and $NO_2$. The second set of experiments was conducted under
irradiation of UV lights (300-500 nm, see Appendix A for irradiation spectrum). Under such
conditions, NO, $NO_2$ and $O_3$ reached photochemical steady state, which combined the isotopic
effects of EIE, KIE and PHIFE.

In all experiments, the concentrations of NO, $NO_2$ and $O_3$ were allowed to reach steady

state, and the product $NO_2$ was collected from the chamber using a honeycomb denuder tube. After
the NO, $NO_2$ and $O_3$ concentrations reached steady-state, well-mixed chamber air was drawn out
through a 40 cm long Norprene Thermoplastic tubing at 10 L $min^{-1}$ and passed through a
honeycomb denuder system (Chemcomb 3500, Thermo Scientific). Based on flow rate, the $NO_2$
residence time in the was less than 0.5 second, thus in the light-on experiments where NO and $O_3$
coexisted, the $NO_2$ produced inside the transfer tube through $NO+O_3$ reactions should be <0.03
nmol $mol^{-1}$ (using the upper limit of NO and $O_3$ concentrations in our experiments). The
honeycomb denuder system consisted of two honeycomb denuder tubes connected in series. Each
honeycomb denuder tube is a glass cylinder of 38 mm long, 47 mm in diameter, and consist of 212
hexagonal tubes with inner diameters of 2 mm. Before collecting samples, each denuder tube was
coated with a solution of 10% KOH and 25% guaiacol in methanol and then dried by flowing $N_2$
gas through the denuder tube for 15 seconds (Williams and Grosjean, 1990, Walters et al., 2016).
The $NO_2$ reacted with guaiacol coating and was converted into $NO_2^-$ that was retained on the
denuder tube wall (Williams and Grosjean, 1990). NO was inert to the denuder tube coating: a
control experiment sampled pure NO using the denuder tubes, which did not show any measurable
$NO_2^-$. The $NO_2$ collection efficiency of a single honeycomb denuder tube was tested in another
control experiment: air containing 66 nmol mol$^{-1}$ of $NO_2$ was drawn out of the chamber through a
denuder tube, and the $NO_2$ concentration at the exit of the tube holder was measured and found to
be below the detection limit (<1 nmol mol$^{-1}$), suggesting the collection efficiency was nearly 100%
when $[NO_2]$ <66 nmol mol$^{-1}$. Furthermore, when the denuder system consisted of two denuder
tubes in series and $NO_2^-$ in the second denuder was below the detection limit indicating trivial $NO_2$
breakthrough. Each $NO_2$ collection lasted for 0.5-3 hours in order to collect enough $NO_2^-$ for
isotopic analysis (at least 300 nmol). After collection, the $NO_2^-$ was leached from each denuder
tube by rinsing thoroughly with 10 ml deionized water into a clean polypropylene container and
stored frozen until isotopic analysis. Isotopic analysis was conducted at Purdue Stable Isotope
Laboratory. For each sample, approximately 50 nmol of the $NO_2^-$ extract was mixed with 2 M
sodium azide solution in acetic acid buffer in an air-tight glass vial, then shaken overnight to
completely reduce all the $NO_2^-$ to $N_2O_{(g)}$ (Casciotti & McIlvin, 2007; McIlvin & Altabet, 2005).
The product $N_2O$ was directed into a Thermo GasBench equipped with cryo-trap, then the $\delta^{15}N$ of
the $N_2O$ was measured using a Delta-V Isotope Ratios Mass Spectrometer. Six coated denuders
tubes that did not get exposed to $NO_2$ were also analyzed using the same chemical procedure,
which did not show any measurable signal on the IRMS, suggesting the blank from both sampling
process and the chemical conversion process was negligible. The overall analytical uncertainty for
$\delta^{15}N$ analysis was 0.5 ‰ (1$\sigma$) based on replicate analysis of in house $NO_2^-$ standards.

## 3. Results and Discussions

### 3.1. Equilibrium Isotopic Fractionation between NO and NO$_2$

The equilibrium isotope fractionation factor, $\alpha$(NO$_2$-NO), is the $^{15}$N enrichment in NO$_2$ relative to NO, and is expressed as the ratio of rate constants $k_2 / k_1$ of two reactions:

$$^{15}NO_2 + {}^{14}NO \rightarrow {}^{15}NO + {}^{14}NO_2 \qquad\qquad \text{R1, rate constant} = k_1$$

$$^{15}NO + {}^{14}NO_2 \rightarrow {}^{15}NO_2 + {}^{14}NO \qquad\qquad \text{R2, rate constant} = k_2 = k_1\,\alpha(NO_2\text{-}NO)$$

where $k_1$ is the rate constant of the isotopic exchange, which was previously determined to be $8.14 \times 10^{-14}$ cm$^3$ s$^{-1}$ (Sharma et al., 1970). The reaction time required for NO-NO$_2$ to reach isotopic equilibrium was estimated using the exchange rate constants in a simple kinetics box model (BOXMOX, Knote et al., 2015). The model predicts that at typical NO$_x$ concentrations used during the chamber experiments (7.7-62.4 nmol mol$^{-1}$), isotopic equilibrium would be reached within 15 minutes (see Appendix B). Since the sample collection usually started 1 hour after NO$_x$ was well mixed in the chamber, there was sufficient time to reach full isotope equilibrium. The isotope equilibrium fractionation factor ($\alpha$(NO$_2$-NO)) is then calculated to be:

$$\alpha(NO_2 - NO) = \frac{[^{15}NO_2][^{14}NO]}{[^{14}NO_2][^{15}NO]} = \frac{R(NO_2)}{R(NO)} \qquad\qquad \text{Eq. (1)}$$

where R(NO, NO$_2$) are the $^{15}$N/$^{14}$N ratios of NO and NO$_2$. By definition, the $\delta^{15}$N(NO)=(R(NO)/R(reference) -1)$\times$1000 ‰ and $\delta^{15}$N(NO$_2$)=(R(NO$_2$)/R(reference)-1) $\times$1000 ‰, but hereafter, the $\delta^{15}$N values of NO, NO$_2$ and NO$_x$ will be referred as $\delta$(NO), $\delta$(NO$_2$) and $\delta$(NO$_x$), respectively. Eq. (1) leads to:

$$\delta(NO_2) - \delta(NO) = (\alpha(NO_2 - NO) - 1)(1 + \delta(NO)) \qquad\qquad \text{Eq. (2)}$$

Using Eq. (2) and applying NO$_x$ isotopic mass balance ($\delta$(NO$_x$)=$f$(NO$_2$)$\delta$(NO$_2$)+(1-$f$(NO$_2$))$\delta$(NO), $f$(NO$_2$)=[NO$_2$]/([NO]+[NO$_2$])) yields:

$$\frac{\delta(NO_2) - \delta(NO_x)}{1 + \delta(NO_2)} = \frac{\alpha(NO_2 - NO) - 1}{\alpha(NO_2 - NO)}\left(1 - f(NO_2)\right) \qquad\qquad \text{Eq. (3)}$$

Here, $\delta(NO_x)$ equals to the $\delta^{15}N$ value of the cylinder NO and $f(NO_2)$ is the molar fraction of $NO_2$
with respect to total $NO_x$. Three experiments (Table 1) that measured $\delta(NO_x)$ showed consistent
$\delta(NO_x)$ values of (-58.7±0.8) ‰ (n = 3), indicating $\delta(NO_x)$ remained unchanged throughout the
experiments (as expected for isotope mass balance). Thus, the $\delta(NO_x)$ can be treated as a constant
in Eq. (3), and the linear regression of $(\delta(NO_2)-\delta(NO_x))/(1+\delta(NO_2))$ versus $1-f(NO_2)$ should have
an intercept of 0 and a slope of $(\alpha(NO_2\text{-}NO)\text{-}1)/\alpha(NO_2\text{-}NO)$.

The plot of $(\delta(NO_2)-\delta(NO_x))/(1+\delta(NO_2))$ as a function of $1-f(NO_2)$ values from five

experiments yields an $\alpha(NO_2\text{-}NO)$ value of 1.0289±0.0019 at room temperature (Fig. 1B and Table
1). This fractionation factor is comparable to previously measured values but with some
differences. Our result agrees well with the $\alpha(NO_2\text{-}NO)$ value of 1.028±0.002 obtained by Begun
and Melton (1956) at room temperature. However, Walters et al., (2016) determined the $\alpha(NO_2\text{-}$
NO) values of $NO\text{-}NO_2$ exchange in a 1-liter reaction vessel, which showed a slightly higher
$\alpha(NO_2\text{-}NO)$ value of 1.035. This discrepancy might originate from rapid heterogeneous reactions
on the wall of the reaction vessel at high $NO_x$ concentrations and the small chamber size used by
Walters et al. (2016). They used a reaction vessel made of Pyrex, which is known to absorb water
(Do Remus et al., 1983; Takei et al., 1997) that can react with $NO_2$ forming HONO, $HNO_3$ and
other N compounds. Additionally, previous studies have suggested that Pyrex walls enhance the
formation rate of $N_2O_4$ by over an order of magnitude (Barney & Finlayson-Pitts, 2000; Saliba et
al., 2001), which at isotopic equilibrium is enriched in $^{15}N$ compared to NO and $NO_2$ (Walters &
Michalski, 2015). Therefore, their measured $\alpha(NO_2\text{-}NO)$ might be slightly higher than the actual
$\alpha(NO_2\text{-}NO)$ value. In this work, the 10 m$^3$ chamber has a much smaller surface to volume ratio
relative to Walters et al. (2016) which minimizes wall effects, and the walls were made of Teflon
that minimize $NO_2$ surface reactivity, which was evidenced by the $NO_2$ wall loss control
experiment. Furthermore, the low $NO_x$ mixing ratios in our experiments minimized $N_2O_4$ and $N_2O_3$
formation. At NO and $NO_2$ concentrations of 50 nmol mol$^{-1}$ the steady state concentrations of $N_2O_4$
and $N_2O_3$ were calculated to be 0.014 and 0.001 pmol mol$^{-1}$, respectively (Atkinson et al., 2004).
Therefore, we suggest our measured $\alpha(NO_2$-NO$)$ value (1.0289±0.0019) may better reflect the
room temperature (298 K) $NO$-$NO_2$ EIE in the ambient environment.

Unfortunately, the chamber temperature could not be controlled so we were not able to

investigate the temperature dependence of the EIE. Hence, we speculate that the $\alpha(NO_2$-NO$)$
follows a similar temperature dependence pattern calculated in Walters et al. (2016). Walters et al.
(2016) suggested that, the $\alpha(NO_2$-NO$)$ value would be 0.0047 higher at 273 K and 0.002 lower at
310 K, relative to room temperature (298 K). Using this pattern and our experimentally determined
data, we suggest the $\alpha(NO_2$-NO$)$ values at 273 K, 298 K and 310 K are 1.0336±0.0019,
1.0289±0.0019 and 1.0269±0.0019, respectively. This 0.0067 variation at least partially contribute
to the daily and seasonal variations of $\delta^{15}N$ values of $NO_2$ and nitrate in some areas (e.g., polar
regions with strong seasonal temperature variation). Thus, future investigations should be
conducted to verify the EIE temperature dependence.

**3.2. Kinetic isotopic fractionation of Leighton Cycle**

The photochemical reactions of $NO_x$ will compete with the isotope exchange fractionations

between NO and $NO_2$. The $NO$-$NO_2$ photochemical cycle in the chamber was controlled by the
Leighton cycle: $NO_2$ photolysis and the $NO + O_3$ reaction. This is because there were no VOCs in
the chamber so no $RO_2$ was produced, which excludes the $NO + RO_2$ reaction. Likewise, the low
water vapor content (RH<10%) and the minor flux of photons < 310 nm results in minimal OH
production and hence little $HO_2$ formation and subsequently trivial amount of $NO_2$ would be
formed by NO + HO$_2$. Applying these limiting assumptions, the EIE between NO and NO$_2$ (R1-
R2) were only competing with the KIE (R3-R4) and the PHIFE in R5-R6:

$^{14}NO_2 \rightarrow {}^{14}NO+O$                                        R3, rate constant=$j(NO_2)$

$^{15}NO_2 \rightarrow {}^{15}NO+O$                                        R4, rate constant=$j(NO_2)\ \alpha_1$

$^{14}NO+O_3 \rightarrow {}^{14}NO_2+O_2$                              R5, rate constant=$k_5$

$^{15}NO+O_3 \rightarrow {}^{15}NO_2+O_2$                              R6, rate constant=$k_5\ \alpha_2$

In which $j(NO_2)$ is the NO$_2$ photolysis rate ($1.4\times10^{-3}$ s$^{-1}$ in these experiments), $k_5$ is the rate constant
for the NO+O$_3$ reaction ($1.73\times10^{-14}$ cm$^3$ s$^{-1}$, Atkinson et al., 2004), and $\alpha_{1,2}$ are isotopic
fractionation factors for the two reactions. Previous studies (Freyer et al., 1993; Walters et al.,
2016) have attempted to assess the competition between EIE (R1-R2), KIE and PHIFE (R3-R6),
but none of them quantified the relative importance of the two processes, nor were $\alpha_1$ or $\alpha_2$ values
experimentally determined. Here we provide the mathematical solution of EIE, KIE and PHIFE to
illustrate how R1-R6 affect the isotopic fractionations between NO and NO$_2$.

First, the NO$_2$ lifetime with respect to isotopic exchange with NO ($\tau_{exchange}$) and photolysis

($\tau_{photo}$) was determined:

$$\tau_{exchange} = \frac{1}{k_1\,[NO]}$$                                   Eq. (4)

$$\tau_{photo} = \frac{1}{j(NO_2)}$$                                   Eq. (5)

We then define an A factor:

$$A = \begin{cases} \dfrac{\tau_{exchange}}{\tau_{photo}} & \text{when } j(NO_2) \neq 0 \\ 0 & \text{when } j(NO_2) = 0 \end{cases}$$        Eq. (6)

Using R1-R6 and Eq. (1)-(6), we solved steady-state $\delta(NO_2)$ and $\delta(NO)$ values (see calculations
in Appendix C). Our calculations show that the $\delta(NO_2)$-$\delta(NO)$ and $\delta(NO_2)$-$\delta(NO_x)$ values at steady
state can be expressed as functions of $\alpha_1$, $\alpha_2$, $\alpha(NO_2\text{-}NO)$ and A:
$\delta(NO_2) - \delta(NO) = \frac{(\alpha_2-\alpha_1)\,A+(\alpha(NO_2-NO)-1)}{\alpha_2 A+\alpha(NO_2-NO)}\left(1 + \delta(NO_2)\right)$
$\approx \frac{(\alpha_2-\alpha_1)\,A+(\alpha(NO_2-NO)-1)}{A+1}\left(1 + \delta(NO_2)\right)$                Eq. (7)
$\delta(NO_2) - \delta(NO_x) = \frac{(\alpha_2-\alpha_1)\,A+(\alpha(NO_2-NO)-1)}{\alpha_2 A+\alpha(NO_2-NO)}\left(1 + \delta(NO_2)\right)\left(1 - f(NO_2)\right)$
$\approx \frac{(\alpha_2-\alpha_1)\,A+(\alpha(NO_2-NO)-1)}{A+1}\left(1 + \delta(NO_2)\right)\left(1 - f(NO_2)\right)$                Eq. (8)
Equation (7) shows the isotopic fractionation between NO and $NO_2$ ($\delta(NO_2)$-$\delta(NO)$) is mainly
determined by A, the EIE factor ($\alpha(NO_2\text{-}NO)$-1) and the ($\alpha_2$-$\alpha_1$) factor assuming ($1+\delta(NO_2)$) is
close to 1. This ($\alpha_2$-$\alpha_1$) represents a combination of KIE and PHIFE, suggesting they act together
as one factor; therefore, we name the ($\alpha_2$-$\alpha_1$) factor Leighton Cycle Isotopic Effect, i.e., LCIE.
Using measured $\delta(NO_2)$, $\delta(NO_x)$ values, A values (Table 1), and the previously determined $\alpha(NO_2\text{-}$
$NO$) value, We plot $\frac{\delta(NO_2)-\delta(NO_x)}{(1+\delta(NO_2))(1-f(NO_2))}$ (equals to $\frac{\delta(NO_2)-\delta(NO)}{(1+\delta(NO_2))}$) against A value and use Equations
(7) and (8) to estimate the ($\alpha_2$-$\alpha_1$) value (Fig. 1C). The plot shows that the best fit for the LCIE
factor is (-10±5) ‰ (Rooted Mean Square Error, RMSE, was lowest when $\alpha_2$-$\alpha_1$ =-10‰). The
uncertainties in the LCIE factor are relatively higher than that of the EIE factor, mainly because
of the accumulated analytical uncertainties at low $NO_x$ and $O_3$ concentrations, and low A values
(0.10-0.28) due to the relatively low $j(NO_2)$ value ($1.4\times10^{-3}$ $s^{-1}$) under the chamber irradiation
conditions.

This LCIE factor determined in our experiments is in good agreement with theoretical

calculations. Walters and Michalski (2016) previously used an *ab initio* approach to determine an
$\alpha_2$ value of 0.9933 at room temperature, 0.9943 at 237 K and 0.9929 at 310 K. The total variation
of $\alpha_2$ values from 273 K to 310 K is only 1.4 ‰, significantly smaller than our experimental
uncertainty (±5 ‰). The $\alpha_1$ value was calculated using a ZPE shift model (Miller & Yung, 2000)
to calculate the isotopic fractionation of $NO_2$ by photolysis. Briefly, this model assumes both
isotopologues have the same quantum yield function and the PHIFE was only caused by the
differences in the $^{15}NO_2$ and $^{14}NO_2$ absorption cross-section as a function of wavelength, thus $\alpha_1$
values do not vary by temperature. The $^{15}NO_2$ absorption cross-section was calculated by shifting
the $^{14}NO_2$ absorption cross-section by the $^{15}NO_2$ zero-point energy (Michalski et al., 2004). When
the ZPE shift model was used with the irradiation spectrum of the chamber lights, the resulting $\alpha_1$
value was 1.0023. Therefore, the theoretically predicted $\alpha_2$-$\alpha_1$ value should be -0.0090, i.e., (-
9.0±0.7) ‰ when temperature ranges from 273 K to 310 K. This result shows excellent agreement
with our experimentally determined room temperature $\alpha_2$-$\alpha_1$ value of (-10±5) ‰.
This model was then used to evaluate the variations of $\alpha_1$ value to different lighting
conditions. The TUV model (TUV5.3.2, Madronich & Flocke, 1999) was used to calculate the
solar wavelength spectrum at three different conditions: early morning/late afternoon (solar zenith
angle=85 degree), mid-morning/afternoon (solar zenith angle=45 degree), noon (solar zenith
angle=0 degree). These spectrums were used in the ZPE shift model to calculate the $\alpha_1$ values,
which are 1.0025, 1.0028, and 1.0029 at solar zenith angles of 85, 45 and 0 degree, respectively.
These values, along with the predicted $\alpha_1$ value in the chamber, showed a total span of 0.6‰
(1.0026±0.0003), which is again significantly smaller than our measured uncertainty. Therefore,
we suggest that our experimentally determined LCIE factor ((-10±5) ‰) can be used in most
tropospheric solar irradiation spectrums.

The equations can also be applied in tropospheric environments to calculate the combined

isotopic fractionations of EIE and LCIE for NO and $NO_2$. First, the $NO_2$ sink reactions (mainly
$NO_2$+OH in the daytime) are at least 2-3 orders of magnitude slower than the Leighton cycle and
the NO-$NO_2$ isotope exchange reactions (Walters et al., 2016), therefore their effects on the $\delta(NO_2)$
should be minor. Second, although the conversion of NO into $NO_2$ in the ambient environment is
also controlled by NO + $RO_2$ and $HO_2$ in addition to NO+$O_3$ (e.g., King et al., 2001), Eq. (7) still
showed good agreement with field observations in previous studies. Freyer et al. (1993)
determined the annual average daytime $\delta(NO_2)$-$\delta(NO)$ at Julich, Germany along with average
daytime NO concentration (9 nmol $mol^{-1}$, similar to our experimental conditions) to be
(+18.03±0.98) ‰. Using Eq. (7), assuming the daytime average $j(NO_2)$ value throughout the year
was (5.0±1.0)×$10^{-3}$, and a calculated A value from measured $NO_x$ concentration ranged from 0.22-
0.33, the average NO-$NO_2$ fractionation factor was calculated to be (+19.8±1.4) ‰ (Fig. 1C), in
excellent agreement with the measurements in the present study. This agreement suggests the
NO+$RO_2$/$HO_2$ reactions might have similar fractionation factors as NO+$O_3$. Therefore, we suggest
Eq. (7) and (8) can be used to estimate the isotopic fractionations between NO and $NO_2$ in the
troposphere.

**3.3 Calculating nitrogen isotopic fractionations of NO-$NO_2$**

First, Eq. (7) was used to calculate the $\Delta(NO_2-NO)$ = $\delta(NO_2)$-$\delta(NO)$ at a wide range of

$NO_x$ concentrations, $f(NO_2)$ and $j(NO_2)$ values (Fig. 2A-D), assuming (1+$\delta(NO_2)$) ≈1. $j(NO_2)$
values of 0 $s^{-1}$ (Fig. 2A), 1.4×$10^{-3}$ $s^{-1}$ (Fig. 2B), 5×$10^{-3}$ $s^{-1}$ (Fig. 2C) and 1×$10^{-2}$ $s^{-1}$ (Fig. 2D) were
selected to represent nighttime, dawn (as well as the laboratory conditions of our experiments),
daytime average and noon, respectively. Each panel represented a fixed $j(NO_2)$ value, and the
$\Delta$(NO$_2$-NO) values were calculated as a function of the A value, which was derived from NO$_x$
concentration and $f$(NO$_2$). The A values have a large span, from 0 to 500, depending on the $j$(NO$_2$)
value and the NO concentration. When A=0 ($j$(NO$_2$)=0) and $f$(NO$_2$)<1 (meaning NO-NO$_2$ coexist
and [O$_3$]=0), Eq. (7) and (8) become Eq. (2) and (3), showing the EIE was the sole factor, the
$\Delta$(NO$_2$-NO) values were solely controlled by EIE which has a constant value of +28.9 ‰ at 298K
(Fig. 2A). When $j$(NO$_2$)>0, the calculated $\Delta$(NO$_2$-NO) values showed a wide range from -10.0 ‰
(controlled by LCIE factor: $\alpha_2$-$\alpha_1$=-10 ‰) to +28.9 ‰ (controlled by EIE factor: $\alpha$(NO$_2$-NO)-1 =
+28.9 ‰). Fig. 2B-D display the transition from a LCIE-dominated regime to an EIE-dominated
regime. The LCIE-dominated regime is characterized by low [NO$_x$] (<50 pmol mol$^{-1}$), representing
remote ocean areas and polar regions (Beine et al., 2002; Custard et al., 2015). At this range the A
value can be greater than 200, thus Eq. (7) can be simplified as: $\Delta$(NO$_2$-NO) = $\alpha_2$-$\alpha_1$, suggesting
the LCIE almost exclusively controls the NO-NO$_2$ isotopic fractionation. The $\Delta$(NO$_2$-NO) values
of these regions are predicted to be <0 ‰ during most time of the day and < -5 ‰ at noon. On the
other hand, the EIE-dominated regime was characterized by high [NO$_x$] (>20 nmol mol$^{-1}$) and low
$f$(NO$_2$) (< 0.6), representative of regions with intensive NO emissions, e.g., near roadside or stack
plumes (Clapp & Jenkin, 2001; Kimbrough et al., 2017). In this case, the $\tau_{exchange}$ are relatively
short (10-50 s) compared to the $\tau_{photo}$ (approximately 100 s at noon and 1000 s at dawn), therefore
the A values are small (0.01-0.5). The EIE factor in this regime thus is much more important than
the LCIE factor, resulting in high $\Delta$(NO$_2$-NO) values (>20 ‰). Between the two regimes, both
EIE and LCIE are competitive and therefore it is necessary to use Eq. (7) to quantify the $\Delta$(NO$_2$-
NO) values.
Fig. 2 also implies that changes in the $j$(NO$_2$) value can cause the diurnal variations in
$\Delta$(NO$_2$-NO) values. Changing $j$(NO$_2$) would affect the value of A and consequently the NO-NO$_2$
isotopic fractionations in two ways: 1) changes in $j(NO_2)$ value would change the photolysis
intensity, therefore the $\tau_{photo}$ value; 2) in addition, changes in $j(NO_2)$ value would also alter the
steady state NO concentration, therefore changing the $\tau_{exchange}$ (Fig. 2C). The combined effect of
these two factors on the A value varies along with the atmospheric conditions, and thus needs to
be carefully calculated using $NO_x$ concentration data and atmospheric chemistry models.

We then calculated the differences of $\delta^{15}N$ values between $NO_2$ and total $NO_x$, e.g. $\Delta(NO_2$-

$NO_x) = \delta(NO_2)$-$\delta(NO_x)$ in Fig. 2E-H. Since $\Delta(NO_2$-$NO_x)$ are connected through the observed $\delta^{15}N$
of $NO_2$ (or nitrate) to the $\delta^{15}N$ of $NO_x$ sources, this term might be useful in field studies (e.g.,
Chang et al., 2018; Zong et al., 2017). The calculated $\Delta(NO_2$-$NO_x)$ values (Fig. 2E-H) also showed
a LCIE-dominated regime at low $[NO_x]$ and an EIE-dominated regime at high $[NO_x]$. The $\Delta(NO_2$-
$NO_x)$ values were dampened by the $1$-$f(NO_2)$ factor comparing to $\Delta(NO_2$-$NO)$, as shown in Eq.
(3) and (8): $\Delta(NO_2$-$NO_x) = \Delta(NO_2$-$NO)$ $(1$-$f(NO_2))$. At high $f(NO_2)$ values ($>0.8$), the differences
between $\delta(NO_2)$ and $\delta(NO_x)$ were less than 5 ‰, thus the measured $\delta(NO_2)$ values were similar to
$\delta(NO_x)$, although the isotopic fractionation between NO and $NO_2$ could be noteworthy. Some
ambient environments with significant NO emissions or high $NO_2$ photolysis rates usually have
$f(NO_2)$ values between 0.4-0.8 (Mazzeo et al., 2005; Vicars et al., 2013). In this scenario, the
$\Delta(NO_2$-$NO_x)$ values in Fig. 2F-H showed wide ranges of -4.8 ‰ to +15.6 ‰, -6.0 ‰ to +15.0 ‰,
and -6.3 ‰ to +14.2 ‰ at $j(NO_2)=1.4\times10^{-3}$ $s^{-1}$, $5\times10^{-3}$ $s^{-1}$, $1\times10^{-2}$ $s^{-1}$, respectively. These significant
differences again highlighted the importance of both LCIE and EIE (Eq. (7) and (8)) in calculating
the $\Delta(NO_2$-$NO_x)$. In the following discussion, we assume 1) the $\alpha_1$ value remain constant (see
discussion above), 2) the $NO$+$RO_2$/$HO_2$ reactions have the same fractionation factors ($\alpha_2$) as
$NO$+$O_3$, and 3) both EIE and LCIE do not display significant temperature dependence, then use
Equations (7) and (8) and this laboratory determined LCIE factor (-10 ‰) to calculate the nitrogen
isotopic fractionation between NO and $NO_2$ at various tropospheric atmospheric conditions.

**4. Implications**
The daily variations of $\Delta(NO_2-NO_x)$ values at two roadside $NO_x$ monitoring sites were
predicted to demonstrate the effects of $NO_x$ concentrations to the $NO-NO_2$ isotopic fractionations.
Hourly NO and $NO_2$ concentrations were acquired from a roadside site at Anaheim, CA
(https://www.arb.ca.gov) and an urban site at Evansville, IN (http://idem.tx.sutron.com) on July
25, 2018. The hourly $j(NO_2)$ values output from the TUV model (Madronich & Flocke, 1999) at
these locations was used to calculate the daily variations of $\Delta(NO_2-NO_x)$ values (Fig. 3A, B) by
applying Eq. (8) and assuming $(1+\delta(NO_2)) \approx 1$. Hourly $NO_x$ concentrations were 12-51 nmol mol$^{-1}$
at Anaheim and 9-38 nmol mol$^{-1}$ at Evansville and the $f(NO_2)$ values at both sites did not show
significant daily variations (0.45±0.07 at Anaheim and 0.65±0.08 at Evansville), likely because
the $NO_x$ concentrations were controlled by the high NO emissions from the road (Gao, 2007). The
calculated $\Delta(NO_2-NO_x)$ values using Eq. (8) showed significant diurnal variations. During the
nighttime, the isotopic fractionations were solely controlled by the EIE, the predicted $\Delta(NO_2-NO_x)$
values were (+14.5±2.0) ‰ and (+8.7±2.1) ‰ at Anaheim and Evansville, respectively. During
the daytime, the existence of LCIE lowered the predicted $\Delta(NO_2-NO_x)$ values to (+9.8±1.7) ‰ at
Anaheim and (+3.1±1.5) ‰ at Evansville while the $f(NO_2)$ values at both sites remained similar.
The lowest $\Delta(NO_2-NO_x)$ values for both sites (+7.0 ‰ and +1.7 ‰) occurred around noon when
the $NO_x$ photolysis was the most intense. In contrast, if one neglects the LCIE factor in the daytime,
the $\Delta(NO_2-NO_x)$ values would be (+12.9±1.5) ‰ and (+10.0±1.6) ‰ respectively, an
overestimation of 3.1 ‰ and 6.9 ‰. These discrepancies suggested that the LCIE played an
important role in the NO-NO$_2$ isotopic fractionations and neglecting it could bias the NO$_x$ source
apportionment using $\delta^{15}N$ of NO$_2$ or nitrate.

The role of LCIE was more important in less polluted sites. The $\Delta$(NO$_2$-NO$_x$) values

calculated for a suburban site near San Diego, CA, USA, again using the hourly NO$_x$
concentrations (https://www.arb.ca.gov, Fig. 3C) and $j$(NO$_2$) values calculated from the TUV
model. NO$_x$ concentrations at this site varied from 1 to 9 nmol mol$^{-1}$ and assuming $(1+\delta$(NO$_2$))$\approx$1.
During the nighttime, NO$_x$ was in the form of NO$_2$ ($f$(NO$_2$) = 1) because O$_3$ concentrations were
higher than NO$_x$, thus the $\delta$(NO$_2$) values should be identical to $\delta$(NO$_x$) ($\Delta$(NO$_2$-NO$_x$) = 0). In the
daytime a certain amount of NO was produced by direct NO emission and NO$_2$ photolysis but the
$f$(NO$_2$) was still high (0.73±0.08). Our calculation suggested the daytime $\Delta$(NO$_2$-NO$_x$) values
should be only (+1.3±3.2) ‰ with a lowest value of -1.3 ‰. These $\Delta$(NO$_2$-NO$_x$) values were
similar to the observed and modeled summer daytime $\delta$(NO$_2$) values in West Lafayette, IN
(Walters et al., 2018), which suggest the average daytime $\Delta$(NO$_2$-NO$_x$) values at NO$_x$ = (3.9±1.2)
nmol mol$^{-1}$ should range from +0.1 ‰ to +2.4 ‰. In this regime, we suggest the $\Delta$(NO$_2$-NO$_x$)
values were generally small due to the significant contribution of LCIE and high $f$(NO$_2$).

The LCIE should be the dominant factor controlling the NO-NO$_2$ isotopic fractionation at

remote regions, resulting in a completely different diurnal pattern of $\Delta$(NO$_2$-NO$_x$) compared with
the urban-suburban area. Direct hourly measurements of NO$_x$ at remote sites are rare, thus we used
total NO$_x$ concentration of 50 pmol mol$^{-1}$, daily O$_3$ concentration of 20 nmol mol$^{-1}$ at Summit,
Greenland (Dibb et al., 2002; Hastings et al., 2004; Honrath et al., 1999; Yang et al., 2002), and
assumed $(1+\delta$(NO$_2$)) $\approx$1 and the conversion of NO to NO$_2$ was completely controlled by O$_3$ to
calculate the NO/NO$_2$ ratios. Here the isotopes of NO$_x$ were almost exclusively controlled by the
LCIE due to the high A values (>110). The $\Delta$(NO$_2$-NO$_x$) values displayed a clear diurnal pattern
(Fig. 3D) with highest value of -0.3 ‰ in the "nighttime" (solar zenith angle >85 degree) and
lowest value of -5.0 ‰ in the mid-day. This suggest that the isotopic fractionations between NO
and $NO_2$ were almost completely controlled by LCIE at remote regions, when $NO_x$ concentrations
were <0.1 nmol $mol^{-1}$. However, since the isotopic fractionation factors of nitrate-formation
reactions ($NO_2$+OH, $NO_3$+HC, $N_2O_5$+$H_2O$) are still unknown, more studies are needed to fully
explain the daily and seasonal variations of $\delta(NO_3^-)$ at remote regions.
Nevertheless, our results have a few limitations. First, currently there are very few field
observations that can be used to evaluate our model, therefore, future field observations that
measure the $\delta^{15}N$ values of ambient NO and $NO_2$ should be carried out to test our model. Second,
more work, including theoretical and experimental studies, is needed to investigate the isotope
fractionation factors occurring during the conversion from $NO_x$ to $NO_y$ and nitrate: in the $NO_y$
cycle, EIE (isotopic exchange between $NO_2$, $NO_3$ and $N_2O_5$), KIE (formation of $NO_3$, $N_2O_5$ and
nitrate) and PHIFE (photolysis of $NO_3$, $N_2O_5$, HONO and sometimes nitrate) may also exist and
be relevant for the $\delta^{15}N$ of $HNO_3$ and HONO. In particular, the N isotope fractionation occurring
during the $NO_2$ + OH $\rightarrow$ $HNO_3$ reaction needs investigation. Such studies could help us modeling
the isotopic fractionation between $NO_x$ emission and nitrate, and eventually enable us to analyze
the $\delta^{15}N$ value of $NO_x$ emission by measuring the $\delta^{15}N$ values of nitrate aerosols and nitrate in wet
depositions. Third, our discussion only focuses on the reactive nitrogen chemistry in the
troposphere, however, the nitrogen chemistry in the stratosphere is drastically different from the
tropospheric chemistry, thus future studies are also needed to investigate the isotopic fractionations
in the stratospheric nitrogen chemistry. Last, the temperature dependence of both EIE and LCIE
needs to be carefully investigated, because of the wide range of temperature in both troposphere
and stratosphere. Changes in temperature could alter the isotopic fractionation factors of both EIE
and LCIE, as well as contribute to the seasonality of isotopic fractionations between $NO_x$ and $NO_y$
molecules.

**5. Conclusions**
The effect of $NO_x$ photochemistry on the nitrogen isotopic fractionations between NO and
$NO_2$ was investigated. We first measured the isotopic fractionations between NO and $NO_2$ and
provided mathematical solutions to assess the impact of $NO_x$ level and $NO_2$ photolysis rate ($j(NO_2)$)
to the relative importance of EIE and LCIE. The EIE and LCIE isotope fractionation factors, at
room temperature, were determined to be 1.0289±0.0019 and 0.990±0.005, respectively. These
calculations and measurements can be used to determine the steady state $\Delta(NO_2\text{-}NO)$ and $\Delta(NO_2\text{-}$
$NO_x)$ values at room temperature. Subsequently we applied our equations to polluted, clean and
remote sites to model the daily variations of $\Delta(NO_2\text{-}NO_x)$ values. We found that the $\Delta(NO_2\text{-}NO_x)$
values could vary from over +20 ‰ to less than -5 ‰ depending on the environment: in general,
the role of LCIE becoming more important at low $NO_x$ concentrations, which tend to decrease the
$\Delta(NO_2\text{-}NO_x)$ values. Our work provided a mathematical approach to quantify the nitrogen isotopic
fractionations between NO and $NO_2$ that can be applied to many tropospheric environments, which
could help interpret the measured $\delta^{15}N$ values of $NO_2$ and nitrate in field observation studies.

**Acknowledgement**
We thank NCAR's Advanced Study Program granted to Jianghanyang Li. The National
Center for Atmospheric Research is operated by the University Corporation for Atmospheric
Research, under the sponsorship of the National Science Foundation. We also thank funding
support from Purdue Climate Change Research Center and A. H. Ismail Interdisciplinary Program
Doctoral Research Travel Award granted by Purdue University.
**Data Availability**
Data acquired from this study was deposited at Open Sciences Framework (Li, 2019,
DOI 10.17605/OSF.IO/JW8HU).
**Author contribution**
J. Li and G. Michalski designed the experiments, X. Zhang and J. Li conducted the
experiments. X. Zhang, G. Michalski, J. Orlando and G. Tyndall helped J. Li in interpreting the
results. The manuscript was written by J. Li and all the authors have contributed during the revision
of this manuscript.
**Competing interest**
The authors declare no competing interest.

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

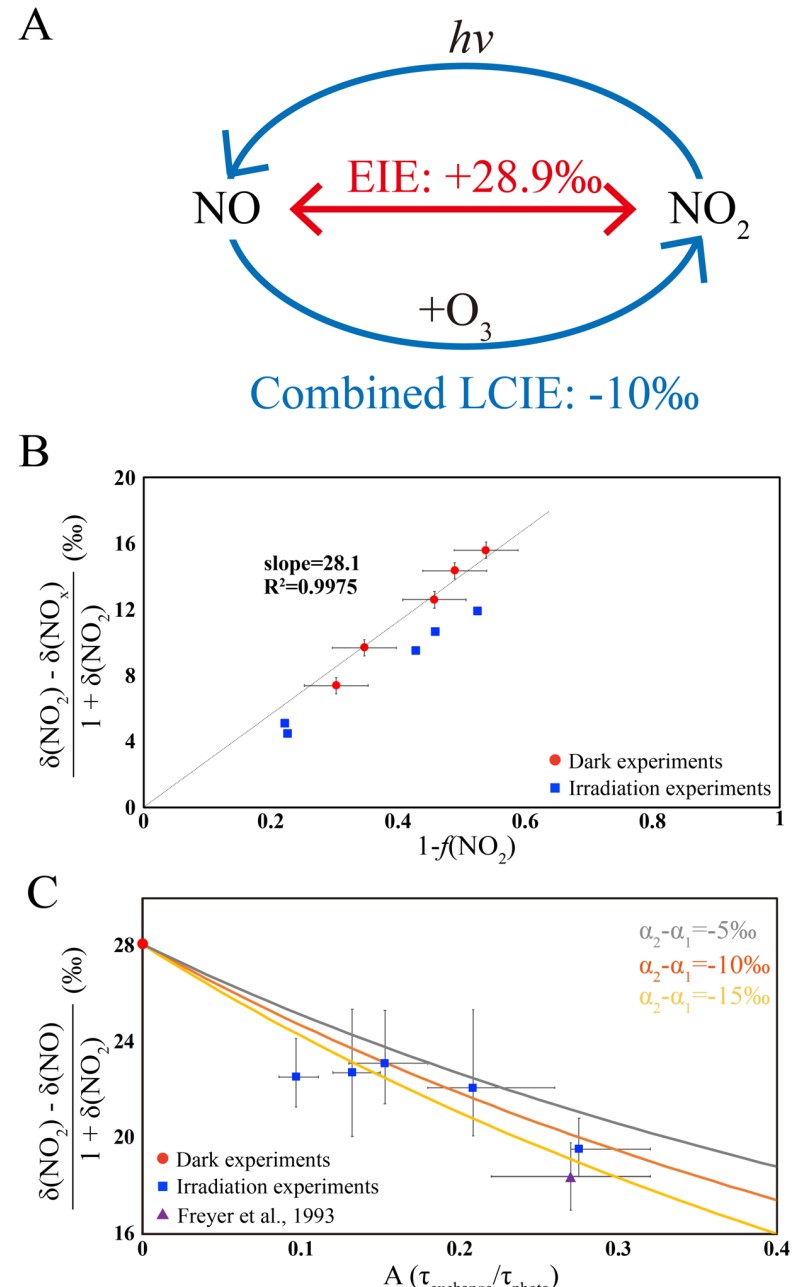

**Fig. 1 A.** a sketch of the isotopic fractionation processes between NO and $NO_2$, both fractionation
factors are determined in this work. **B.** Results from five dark experiments (red circles) yielded a
line with slope of 28.1‰ and an $\alpha(NO_2-NO)$ value of 1.0289, while the results from five UV
irradiation experiments (blue squares) showed a smaller slope; **C.** Results from five UV irradiation
experiments (blue squares) and a previous field study (purple triangle), comparing to the dark
experiments (red circle). The three lines represent different $(\alpha_2-\alpha_1)$ values: the $(\alpha_2-\alpha_1) = -10$ ‰ line
showed the lowest RMSE to our experimental data as well as the previous field observations. The
error bars in panels B and C represented the combined uncertainties of $NO_x$ concentration
measurements and isotopic analysis.

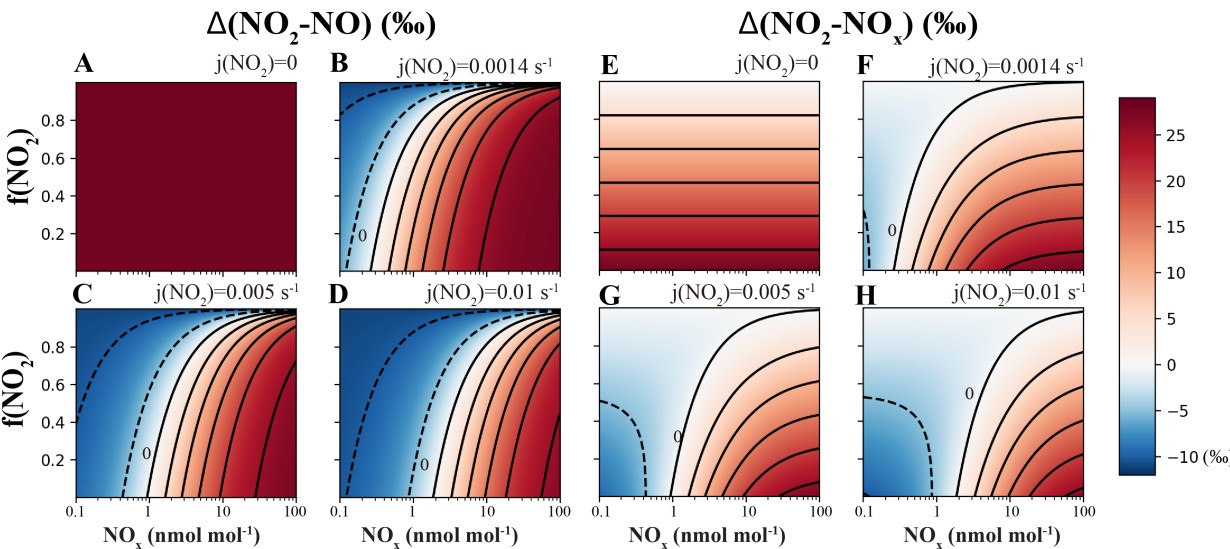

**Fig. 2** Calculating isotopic fractionation values between NO-NO$_2$ ($\Delta$(NO$_2$-NO), **A-D**) and NO$_x$-NO$_2$ ($\Delta$(NO$_2$-NO$_x$), **E-H**) at various $j$(NO$_2$), NO$_x$ level and $f$(NO$_2$) using Eq. (7) and (8). Each panel represents a fixed $j$(NO$_2$) value (showing on the upper right side of each panel), and the fractionation values are shown by color. Lines are contours with the same fractionation values, at an interval of 5‰, the contour line representing 0‰ was marked on each panel except for A and E.

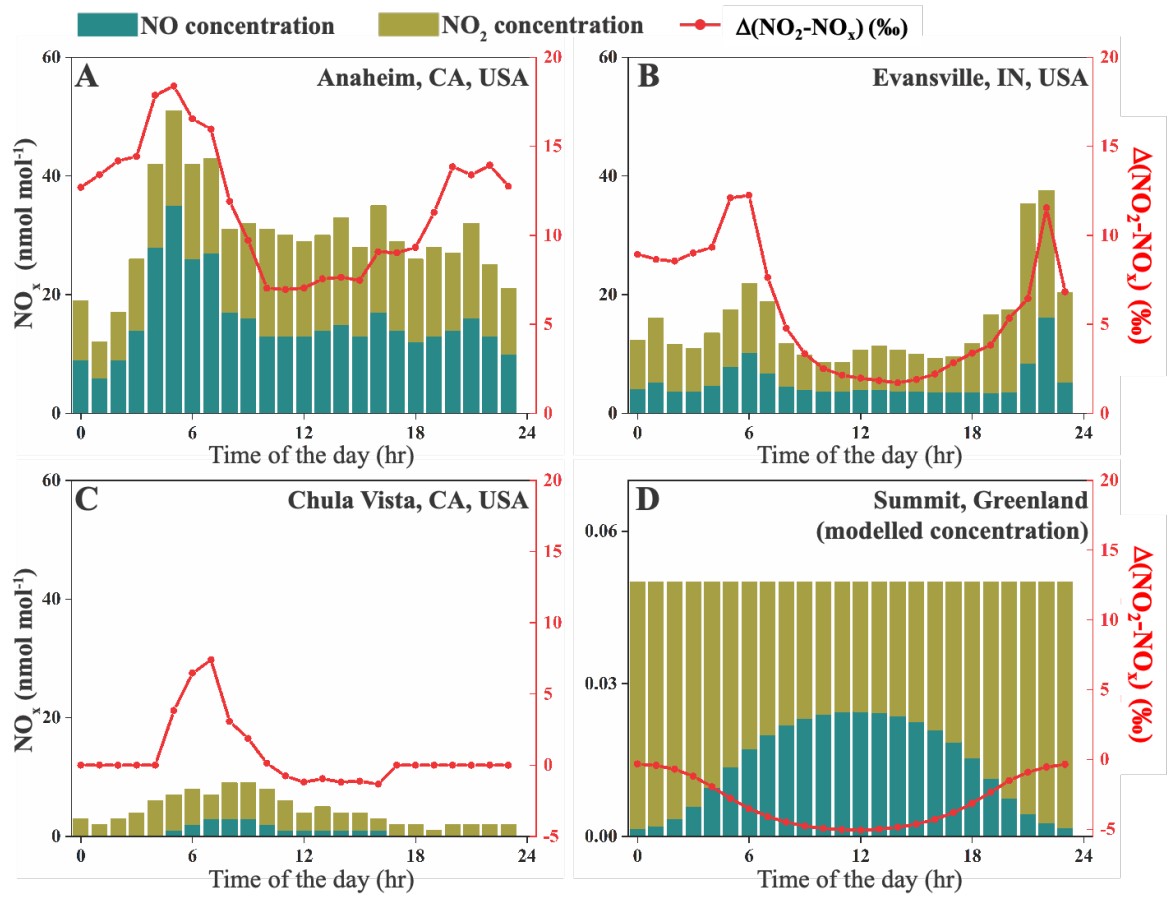

**Fig. 3** $NO_x$ concentrations and calculated $\Delta(NO_2\text{-}NO_x)$ values at four sites. Stacked bars show the
NO and $NO_2$ concentrations extracted from monitoring sites (A-C) or calculated using 0-D box
model (D); the red lines are $\Delta(NO_2\text{-}NO_x)$ values at each site. Note that the $NO_x$ concentration (left-
y) axis on panel D is different from the rest.

| Experiment | Number | NO conc. (ppb) | $NO_2$ conc. (ppb) | $O_3$ conc. (ppb) | $\delta(NO_2)$ (‰) | $f(NO_2)$ |
|---|---|---|---|---|---|---|
| Determining $\delta(NO_x)$ | 1 | 0.0 | 17.8 | 13.4 | -59.5 | 1.00 |
| | 2 | 0.0 | 61.3 | 0.5 | -58.9 | 1.00 |
| | 3 | 0.0 | 18.9 | 10.7 | -58.0 | 1.00 |
| Dark experiments | 1 | 16.0 | 36.8 | 0.0 | -51.8 | 0.70 |
| | 2 | 33.6 | 28.8 | 0.0 | -43.9 | 0.46 |
| | 3 | 6.7 | 12.6 | 0.0 | -49.6 | 0.65 |
| | 4 | 16.2 | 16.9 | 0.0 | -45.1 | 0.51 |
| | 5 | 20.4 | 24.2 | 0.0 | -46.8 | 0.54 |
| Irradiation experiments | 1 | 7.1 | 6.4 | 2.8 | -47.5 | 0.47 |
| | 2 | 4.5 | 5.3 | 4.5 | -48.7 | 0.54 |
| | 3 | 3.3 | 4.4 | 4.2 | -49.8 | 0.57 |
| | 4 | 2.5 | 8.5 | 10.7 | -54.6 | 0.77 |
| | 5 | 5.2 | 18.1 | 11.0 | -54.0 | 0.78 |

Table 1. Experimental conditions, concentrations of NO, $NO_2$ and $O_3$ at steady state, and measured
$\delta(NO_2)$ values.

**Appendix A. Chamber descriptions**
The chamber is a 10 m$^3$ Teflon bag equipped with several standard instruments including
temperature and humidity probe, $NO_x$ monitor and $O_3$ monitor. 128 wall-mounted blacklight tubes
surrounded the chamber to mimic tropospheric photochemistry and the photolysis rate of $NO_2$
($j(NO_2)$) when all lights are on have been previously determined to be $1.4\times10^{-3}$ s$^{-1}$, similar to a
$j(NO_2)$ coefficient at an 81-degree solar zenith angle. The irradiation spectrum of the blacklights
are shown in Figure A1. The chamber was kept at room temperature and one atmospheric pressure.
Before each experiment, the chamber was flushed with zero air at 40 L min$^{-1}$ for at least 12 hours
to ensure the background $NO_x$, $O_3$ and other trace gases were below detection limit.

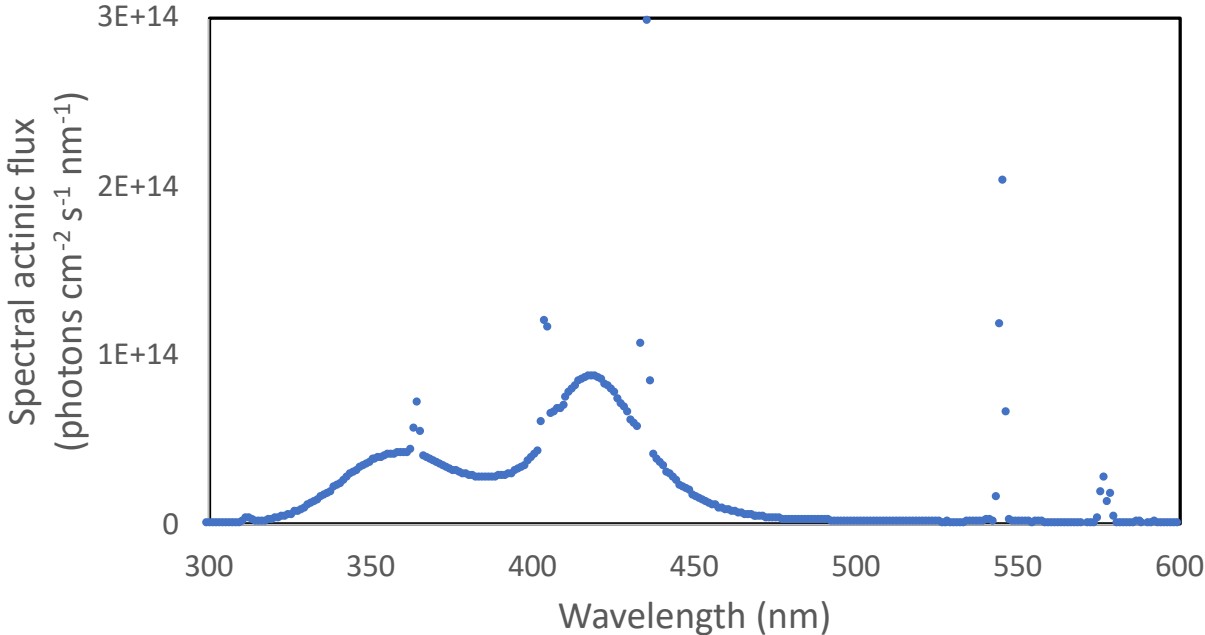


Figure A1 Spectral actinic flux versus wavelengths of the UV light source used in our experiments.

**Appendix B. Box model assessing the time needed for NO-NO₂ to reach isotopic equilibrium**


The time needed to reach NO-NO$_2$ isotopic equilibrium during light-off experiments were
assessed using a 0-D box model. This box model contains only two reactions:
$^{15}NO_2+^{14}NO \rightarrow {}^{15}NO+^{14}NO_2$        $k=8.14000 \times 10^{-14}$ cm$^3$ s$^{-1}$
$^{15}NO+^{14}NO_2 \rightarrow {}^{15}NO_2+^{14}NO$        $k'=8.37525 \times 10^{-14}$ cm$^3$ s$^{-1}$
Where k and k' are rate constants of the reactions. The differences in rate constants were calculated
by assuming an $\alpha$(NO$_2$-NO) value of 1.0289. Six simulations were conducted at various initial NO
(with $\delta^{15}N=0$‰) and O$_3$ levels that were similar to our experiment. Then the $\delta^{15}N$ values of NO
and NO$_2$ during the simulation were calculated from the model and were shown in Figure B1,
suggesting that in our experimental condition, all systems should reach isotopic equilibrium within
1 hr.

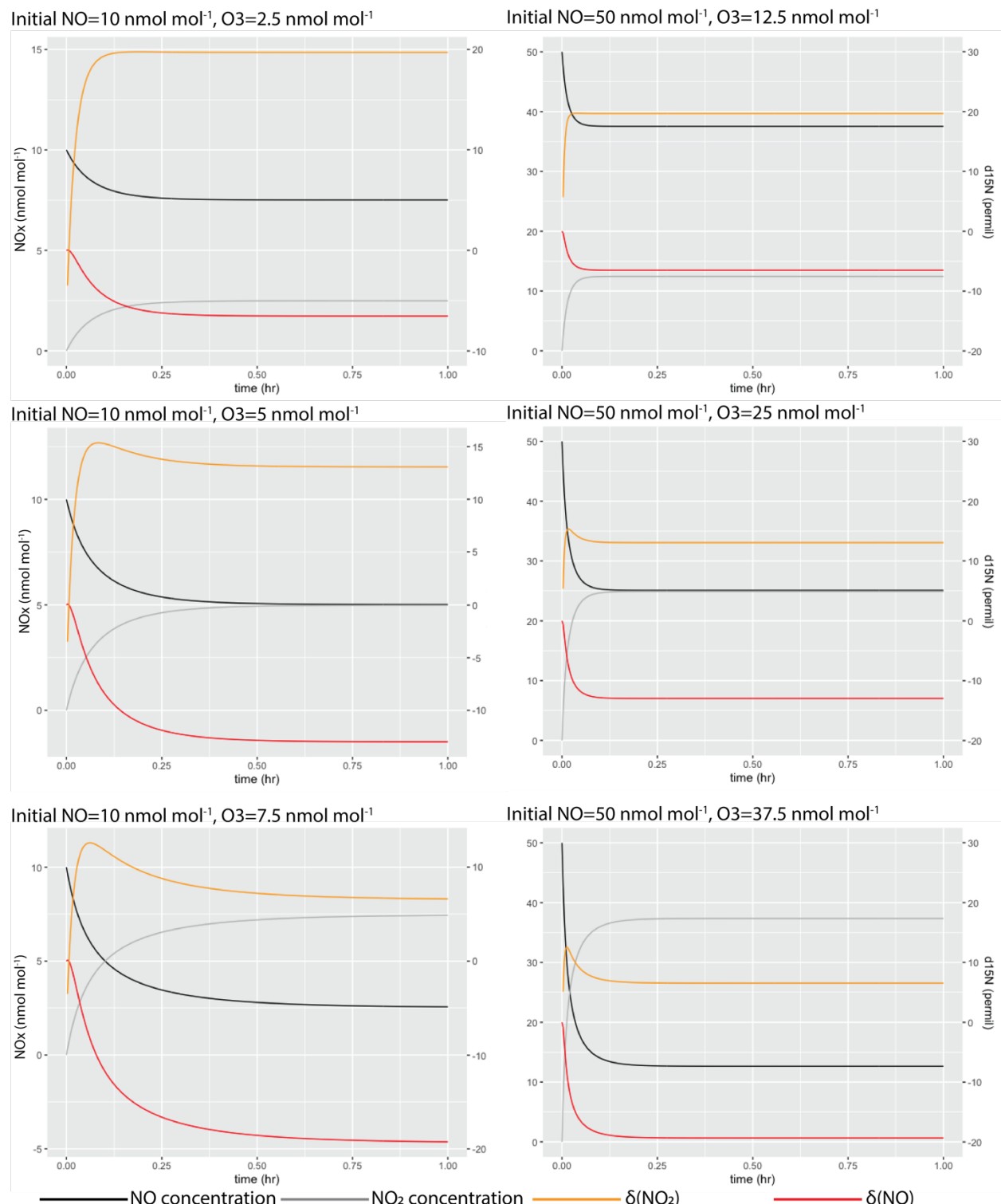


Figure B1 Simulated NO-NO$_2$ isotopic equilibrium process in the chamber at various NO and O$_3$

concentrations.

**Appendix C. Deriving Equations 7 and 8**

When the system (R1-R6) reaches steady-state, we have:

$$d[^{15}NO_2]/dt=0 \qquad \text{Eq. (C1)}$$

Therefore, using R1-R6:

$$k_1\,[^{15}NO_2][^{14}NO]+j(NO_2)\alpha_1[^{15}NO_2]=$$

$$k_5\alpha_2[^{15}NO][O_3]+ k_1\alpha(NO_2\text{-}NO)\,[^{15}NO][^{14}NO_2] \qquad \text{Eq. (C2)}$$

From here we refer $^{14}NO_2$ and $^{14}NO$ as $NO_2$ and $NO$ for convenience, rearrange the above equation, we get:

$$\frac{[^{15}NO_2]}{[^{15}NO]} = \frac{k_5\alpha_2[O_3]+k_1\alpha(NO_2-NO)\,[NO_2]}{j_{NO2}\alpha_1+k_1[NO]} \qquad \text{Eq. (C3)}$$

Meantime, since the Leighton cycle reaction still holds for the majority isotopes ($NO$ and $NO_2$), we have:

$$j_{NO2}[NO_2]= k_5[NO][O_3] \qquad \text{Eq. (C4)}$$

Thus,

$$\frac{[NO_2]}{[NO]} = \frac{k_5\times[O_3]}{j_{NO2}} \qquad \text{Eq. (C5)}$$

From the text, when $j_{NO2}>0$, we defined $A=\tau_{exchange}/\tau_{photo}=j_{NO2}/(k_1\times[NO])$. Using the above equations, we know:

$$\frac{j_{NO2}}{[NO]} = \frac{k_5[O_3]}{[NO_2]} = Ak_1 \qquad \text{Eq. (C6)}$$

$$\frac{j_{NO2}}{k_1[NO]} = \frac{k_5[O_3]}{k_1[NO_2]} = A \qquad \text{Eq. (C7)}$$

Next, to calculate $\delta(NO_2)\text{-}\delta(NO)$, we use the definition of delta notation:

$$\delta(NO_2)\text{-}\delta(NO) = R_{NO2}/R_{std}\text{-} R_{NO}/R_{std} = (R_{NO2}/R_{NO}\text{-}1)(1+\delta(NO)) \qquad \text{Eq. (C8)}$$

$$\frac{R_{NO2}}{R_{NO}} = \frac{[^{15}NO_2][NO]}{[^{15}NO][NO_2]} = \frac{k_5\alpha_2[O_3][NO]+k_1\alpha(NO_2-NO)[NO_2][NO]}{j_{NO2}\alpha_1[NO_2]+k_1[NO][NO_2]}$$
Eq. (C9)

Divide both side by $k_1[NO][NO_2]$:
$$\frac{R_{NO2}}{R_{NO}} = \frac{\frac{k_5\alpha_2[O_3]}{k_1[NO_2]}+\alpha(NO_2-NO)}{\frac{j_{NO2}\alpha_1}{k_1[NO]}+1}$$
Eq. (C10)

Rearrange and substitute $\frac{k_5[O_3]}{k_1[NO_2]}$ and $\frac{j_{NO2}}{k_1[NO]}$ with A:
$$\frac{R_{NO2}}{R_{NO}} = \frac{\alpha_2A+\alpha(NO_2-NO)}{\alpha_1A+1}$$
Eq. (C11)

$$\frac{R_{NO}}{R_{NO2}} = \frac{\alpha_1A+1}{\alpha_2A+\alpha(NO_2-NO)}$$
Eq. (C12)

$$\frac{R_{NO}}{R_{NO2}} - 1 = \frac{(\alpha_1-\alpha_2)A-(\alpha(NO_2-NO)-1)}{\alpha_1A+\alpha(NO_2-NO)}$$
Eq. (C13)

Thus,
$$\delta(NO_2)\text{-}\delta(NO)=\frac{(\alpha_2-\alpha_1)A+(\alpha(NO_2-NO)-1)}{\alpha_1A+\alpha(NO_2-NO)}(1+\delta(NO_2))$$
Eq. (C14)

Then, using mass balance:
$$\delta(NO_2)\,f(NO_2)+\delta(NO)(1\text{-}f(NO_2)) = \delta(NO_x)$$
Eq. (C15)

We can derive Eq. 8:
$$\delta(NO_2)\text{-}\delta(NO_x)=\frac{(\alpha_2-\alpha_1)\times A+\alpha(NO_2-NO)-1)}{\alpha_1A+\alpha(NO_2-NO)}(1+\delta(NO_2))(1\text{-}f(NO_2))$$
Eq. (C16)