# Peer review of "Quantifying the nitrogen isotope effects during photochemical"

_Atmospheric Chemistry and Physics, 2019_

## Referee Comment (RC1) · Matthew Johnson (Referee) · 23 Feb 2020

Review of Quantifying the nitrogen equilibrium and photochemistry-induced isotopic effects between NO and NO$_2$ by Li, Zhang, Michalski, Orlando and Tyndall.

This paper is a professional investigation of an important question that yields valuable insight. Chamber experiments were carried out to measure the isotopic fractionation factors of the Equilibrium Isotopic Effect (EIE) and the Leighton Cycle Induced Isotope Effect (LCIE). A model was used to predict the variations that can be expected in field measurements based on the EIE and LCIE. The experiments seem to yield valuable results and my comments mainly concern model validation (limited because of lack

of data), and I am not convinced the LCIE result is perfectly general. I would like to see more discussion regarding how the LCIE will change with the widely variable conditions of the atmosphere including actinic flux spectrum, temperature, pressure, concentrations of $O_3$, $RO_2$ and $HO_2$, etc.

**Experiment:**
The experimental method and analysis seem sound. For example the use of the honeycomb denuder tube to trap $NO_2$ and reaction chamber with hydrophobic/inert teflon walls which will minimize interference from surface adsorbed water.

**Model and Interpretation**
The semi-analytical PHIFE/ZPE model (Miller 2000; Michalski 2004) is discussed as a way of understanding photolytic isotopic fractionation. Please compare the predictions of that theory with the results of this experiment and comment.

The abstract states that the Leighton cycle isotope effect is $0.990 \pm 0.005$ at room temperature. However, this must be for a certain insolation spectrum and concentrations of $O_3$, $HO_2$, $RO_2$? Please include the conditions. How much will the LCIE change with the changes in conditions found in the atmposhere, or can we take this result to be applicable throughout the atmosphere?

It is argued that the atmospheric LCIE is 18.8 per mil based on the experiment and one field measurement. This may perhaps be sufficient for accepting the proposed value, but no attempt is made to discuss the uncertainty of the measurement, and to predict what variations will be seen in the atmosphere with changes in temperature, actinic flux spectrum, and concentrations of $O_3$, $HO_2$, and $RO_2$. Please present a discussion of these factors.

What affect will the formation of PANs/addition of this equilibrim, have on the LCIE?

Please comment on the LCIE that would be observed in the stratosphere.

In the authors' experiment, NO is converted to $NO_2$ by $O_3$ in conditions with low concentrations of $HO_2$ and $RO_2$, which will play a role in the atmosphere. They suggest that the $HO_2$ and $RO_2$ oxidations of NO might have a similar KIE as the $O_3$ oxidation, but this argument could be considered convenient. It would be stronger with experimental evidence and with improved validation by field measurements. Please make sure to discuss the potential uncertainty that is being introduced in transferring the laboratory results to the field.

No description is made of the UV lights that were used for the photolysis, please add this. According to PHIFE/ZPE, photolytic isotopic fractionation changes as a function of wavelength. How did the spectrum of the lamps used differ from the solar actinic flux spectrum? What wavelength dependence do you expect? How will the LCIE change as a function of altitude in the atmosphere as the actinic flux spectrum changes?

There is precious little field data to use to validate the model. Please comment on what studies you would like to see in order to test the model, and as I have noted, please discuss the impact of different environmental factors, other than NOx concentration, on the results.

**Presentation:**
The abstract is rather short given the interesting findings of the paper. Please expand.

I am not sure why the TLA (three letter acronymn) 'EIE' for 'equilibrium isotope effect' is introduced when there is already the widely accepted idea of the exchange reaction. This could make the abstract obscure for non-specialists.

The introduction should include discussion of photolytic re-emission of deposited nitrate.

Please italicize the symbols used for physical quantities such as *f*, *j* and *k*.

Please add a scheme or figure giving an overview of the key reactions involved in this work.

On page 11 line 230, some *j* values (photolysis rates) are presented. Please include

the units with these numbers. Also in figure 2, '$j = 0.005$', but what are the units?

---

## Short Comment (SC1) · 2 Mar 2020

Regarding the link between the atmospheric variation of 15N and that observed by Hastings, 2004 on nitrate in Greenland snow, I question this causal link. It is indeed easy to show that a 1cm layer of snow at a density of 0.1 and a concentration of 10-6 Mol/L (values in the low range) of nitrate is equivalent to a 500m thick atmospheric layer at 50-12 mol/mol of NOx. It is thus not realistic to think that such thick atmosphere can be leached out in one day, especially giving the stratification of the atmosphere on such icy surfaces (the snow is a low-pass filter). The lack of concentration variation in snow observed by Hastings but also by Erbland et al. 2013 for Antarctica (fig 24 supplementary information) do not argue in favor of a direct link between the atmosphere and snow on hourly time-scale. Therefore, either the (necessarily small) amount of nitrogen deposited daily must have an extreme isotopic composition to be able to imprint the snow at this time-scale (as suggested by Hastings in her paper) and therefore incompatible with the fractionation proposed by the authors, or the isotopic variation observed by Hastings is only incidental and is more a reflection of spatial variability or, in other words, of a poorly estimated spatial signal-to-noise ratio than a robust observation.

I will suggest the authors to be very careful with such polar confrontation. In my view the paper is strong enough even without this confrontation. It contains robust quantification, clear delineated domains, and the possibility to test the conclusions.

Hastings, M. G., Steig, E. J., and Sigman, D. M.: Seasonal variations in N and O isotopes of nitrate in snow at Summit, Greenland: Implications for the study of nitrate in snow and ice cores, J. Geophys. Res., 109, D20306, 10.1029/2004jd004991, 2004.

Erbland, J., Vicars, W. C., Savarino, J., Morin, S., Frey, M. M., Frosini, D., Vince, E., and Martins, J. M. F.: Air-snow transfer of nitrate on the East Antarctic Plateau - Part 1: Isotopic evidence for a photolytically driven dynamic equilibrium in summer, Atmos. Chem. Phys., 13, 6403-6419, 10.5194/acp-13-6403-2013, 2013.

**ACPD**

---

## Referee Comment (RC2) · Anonymous Referee #2 · 16 Mar 2020

This work presents a framework for interpreting isotope effects associated with NO and NO2 cycling in the atmosphere. The study utilizes controlled (chamber-based) experiments to quantify equilibrium and Leighton cycle based isotope effects, and then report under what conditions these effects could be important. The results are certainly important and warrant consideration for publication. Quantifying these effects is critical to being able to incorporate the isotopic composition of nitrogen oxides as a tracer in future models.

There are a number of important issues to address in the work. Here are general comments:

[Figure]

First and foremost, how was the d15N of NOx measured (lines 150-151)? This is critical in that these values are used to demonstrate and calculate the observed isotope effects. Second, was the d15N data of all samples corrected for potential isobaric influences of 17O? (lines 120-124). The generated O3 should have a high D17O that will be transferred to the product NO2. This may impact both the starting NO source d15N values and the measured NO2 values during both dark and photochemical experiments. This could cause an important changes in the findings if excess 17O has not been accounted for in correcting the d15N data.

Related to the above, in Figure 1A it appears that the data was forced through an intercept of 0. But the best fit to the data does not appear to go through 0. What is the slope of the data not forced through the intercept? What might the intercept indicate – from my read this could indicate a shift due to the influence of 17O on the 45 signal when quantifying the isotopic ratios from N2O. A change in slope with the best fit might also bring the calculated value to something that is actually closer to the measured value by Walters et al. Wouldn't this be more consistent with current thinking?

For Figure 1B, the LCIE factor is calculated from "the best fit" (line 206). However, the figure makes it appear qualitative rather than quantitative. This calculation/estimation should be shown quantitatively and an r or r2 value should be reported for the fit. It should also be better explained why the point at A~0.1 and A~0.15 do not follow the expected relationship (why does the difference in d15N not change with A?). Also, why does the relationship have to be linear? In addition, the -10 per mil line does not "best fit" the field observation. It is also not clear what the error bars are on each symbol – I don't see how these should be the same/have the same meaning for the field observation versus the chamber data. The field study point should also be clearly cited in the figure caption. Related to this, in the text (line 215-218) it is argued that the experimental values determined are in good agreement with the field study. But the field study represents a very high NOx environment (at times NOx » O3) and the measurements were taken at different times of the year not just at temperature close

to room temperature. So is it possible that the field determined value is showing a temperature dependence relative to the controlled experiments?

Next, how does the EIE measurement compare with theoretical works? What may drive the differences in the experimental values versus theory? Is formation of other products (such as N2O3 or N2O4) in the chamber a concern and could that influence the measured EIE value?

On line 185-189, it is stated that a2 ("alpha 2") is not quantified. However, Walters and Michalski (2016), which is referenced here, does include an ab initio based estimate for this value. And couldn't this value be used to separate out the magnitude of PHIFE (rather than assume it)?

Specific comments: Abstract – The abstract should be written for a more general audience. Be consistent with abbreviations. For example NO and NO2 are not defined and nitrate isn't abbreviated. Also LCIE should be more clearly defined since this is new terminology in this work. The implications of the study should be better highlighted here. How will a mathematic solution for NO-NO2 isotopic fractionation be useful to the atmospheric chemistry community?

Also, the statement that the new solution can be used at any given condition is a stretch since experiments were only conducted at room temperature and the experiments seem to be most relevant for the troposphere and not stratospheric NOx conditions.

Main Text – Line 25-30: The link between NOx and the formation of nitric acid (i.e. nitrate) needs to be more clearly stated. Also the second sentence is a bit awkwardly phrased given that most of the studies did not use NO2 isotopes directly. It may be worth separating out the studies that have used isotopes of nitrate to understand something about NOx versus studies that have looked at NO2 or NOx specifically. Line 38: remove "the" before chemistry Lines 46-54: Please separate this into at least two separate sentences. Lines 61-63: What drives the difference in the theoretical predictions

for this EIE? Lines 63-64: I think it should be pointed out that this was conducted at room temperatures. Also, the error of $\pm 0.001$ is incorrect? Lines 68-69: KIE and PHIFE for the NOx system is limited but you should probably acknowledge the KIE study on NO + O3: Walters and Michalski (2016) Ab initio study of nitrogen and position-specific oxygen kinetic isotope effects in the NO + O3 reaction, J. Chem. Phys. 145, 224307. Lines 70-75: suggest changing this to "...tends to diminish the expression of the equilibrium isotopic fractionation (EIE) between NO and NO2, but both KIE and PHIFE factors at that time were unknown." It seems strange to suggest that assuming no isotope effects (ie 1) yields no isotope effect. Here would be a good place to better detail the Freyer et al work and approach to determining the single fractionation factor. Otherwise line 75 doesn't really make sense to the reader unfamiliar with the details of Freyer's work. Line 83: atmospheric should be "atmospherically" Line 88: change "NOx nitrogen isotopes" to something more correct like isotopic composition of NOx. Line 105: "was" should be were. Line 105 (and later): what range of wavelengths are used in the experiment. This is important to report as you have already stated! Lines 112-115: More details on the capture of NO using denuder tubes should be provided in the text rather than supplement. And the details are not really given in the suppl either. What was the denuder coated with? How is it determined that there was quantitative and accurate collection of the NO isotopes? What is this method based upon? Lines 120-124: Were corrections conducted for D17O interferences? I imagine the generated O3 will have a high D17O that will be proportional transferred to your product NO2. This may impact both your starting NO source d15N value and your measured NO2 values during dark and photochemical experiments. Section 2 overall – were any blanks tested throughout the experiments? Line 129: difference should be "ratio of" correct? Line 140: I don't understand the formatting here with d(15N, NO)? Line 149-150: How was d15N-NOx measured? If this was done for all experiments, why does n=3? and again how will potential 17O isobaric influences affect your quantification of d15N? Lines 156: Where does the error on the 26.8 value come from? This is not represented in the figure. Lines 166-167: Can you prove that formation of these other

products are not important for the experiments performed here? EIE should be able to be precisely predicted by theory – so why is there such a mismatch between the theoretical and the measured values? Figure 1A: It appears that the data was fitted through an intercept of 0, but the data doesn't appear to go through the intercept. What is the slope of the data when not forced through the intercept? What might the intercept of your data indicate? Maybe D17O isobaric influence? Lines 174-176: What exactly were the wavelengths of the blacklight used in the experiments? Lines 186-189: $\alpha2$ value was determined in Walters and Michalski, 2016 ab initio study as referenced above. Lines 191-197: It might be more straightforward if t(exchange) and t(photo) were defined first and then A, etc. Also please better define the purpose of equations 7 and 8. Also shouldn't these equations have epsilon instead of alpha? Line 207-210: The experimental LCIE should be compared with the NO + O3 KIE. Here and on line 225 it feels a bit like the large uncertainty of +/- 50 percent on the -10 value is being ignored! Line 220: Note that the field experiments sometimes only represent NO2 and other times NOx...so the difference between NO and NO2 was not measured, it was determined. Line 237: I do not see how this shown in Figure 2A. Line 261-263: These are not emissions, these are ambient NO2 and NOx. Also, NOx is not emitted – primary emissions are NO and very on occasion diesel engines have been shown to emit NO2 directly. All of the language here needs to be much more precise.

Section 4 overall: This section needs work. It needs to be established why the sites were chosen. And the sites should be clearly labeled as representative of, for instance, urban versus suburban versus remote versus elevated NOx (roadside) or the like. It seems like the point here actually should be to distinguish the effects across the different sites. Why does it not matter that the O3 (and HO2, RO2, etc) concentrations would vary significantly across these sites?

What are we really learning from site A vs B? They are both roadside. Given that there is such limited data to compare the model with, could the authors compare their roadside model to d15N(NO2) data collected by a roadside such as Felix and Elliott,

2014, "Isotopic composition of passively collected nitrogen dioxide emissions: Vehicle, soil, and livestock source signatures", Atmospheric Environment, 92, 359-366?

Also why not directly compare with what the model would predict quantitatively against the Walters 2018 work? And also predict the values for the Freyer work and compare?

Lines 325-327: This conclusion is a bit strange. There is very little local HNO3 at Summit, Greenland. So drawing the conclusion based upon snow work (not atmosphere and snow) and assuming a direct link temporally between d15NO2 and d15NO3- seems a stretch. It might be more useful to look at Jarvis et al. (2009) instead – Jarvis, J. C., M. G. Hastings, E. J. Steig, and S. A. Kunasek (2009), Isotopic ratios in gas-phase HNO3 and snow nitrate at Summit, Greenland, J. Geophys. Res., 114, D17301, doi:10.1029/2009JD012134. Line 329: what kinds of data and future environmental measurements should be conducted to validate this model? Help the community make this a reality!

Supplement – This needs to be re-read and edited – there are a lot of typos. Some comments on the methods in the supplement – I have a number of questions: -what was the flow rate used to calculate the NO2 residence time ("reside" should be residence in the text)? -include more details of the denuder method – what company are these from? What were they coated with? How do you know they are quantitative in collection? Please show the collection efficiency data. And report how many times this was tested. -need to make it more clear where the 1.0268 value comes from – why is this value assumed here? Based upon the "best fit"? which really isn't a best fit (see comments from above). So what happens here if you do not assume a forced zero intercept? -make sure the editor's suggested technical corrections carry through the supplement as well.

---

## Author Comment (AC1) · 2 Apr 2020

Dear Dr. Johnson,

Thank you for the comments and input to our manuscript. Below is a line-by-line response to your comments. We also attached a PDF version of this response.

Model and Interpretation

The semi-analytical PHIFE/ZPE model (Miller 2000; Michalski 2004) is discussed as a way of understanding photolytic isotopic fractionation. Please compare the predictions of that theory with the results of this experiment and comment.

[Figure]

We discussed the previous calculation of the KIE and added some theoretical calculations in our manuscript. The predictions showed good agreement with our observation (KIE=0.9933, PHIFE=1.0023, therefore predicted LCIE=-9‰ our results=-10ïĆś5‰. In the meantime, we did the ZPE calculation using 4 different actinic flux spectrums (lab light, early morning/late afternoon, mid-morning/afternoon, and noon), all of them showed similar PHIFE values ranging from 1.0023 to 1.0029, suggesting the PHIFE do not vary significantly by light.

The abstract states that the Leighton cycle isotope effect is $0.990 \pm 0.005$ at room temperature. However, this must be for a certain insolation spectrum and concentrations of $O_3$, $HO_2$, $RO_2$? Please include the conditions. How much will the LCIE change with the changes in conditions found in the atmosphere, or can we take this result to be applicable throughout the atmosphere?

We pointed out that, our experiments measured the LCIE fractionation factor when $O_3$ solely controls the NOx cycle. However, we suggest this LCIE factor might be used in the ambient environment because it showed good agreement with previous field observations. Nevertheless, future work is needed to confirm our assumption that the isotopic fractionation factor of $NO+RO_2/HO_2$ are similar to that of $NO+O_3$.

It is argued that the atmospheric LCIE is 18.8 per mil based on the experiment and one field measurement. This may perhaps be sufficient for accepting the proposed value, but no attempt is made to discuss the uncertainty of the measurement, and to predict what variations will be seen in the atmosphere with changes in temperature, actinic flux spectrum, and concentrations of $O_3$, $HO_2$, and $RO_2$. Please present a discussion of these factors.

This 18.8 ‰ value in Freyer's work is an annual average daytime value of combined EIE and LCIE, which was determined using all the measurements in a year. In reality, the value significantly varies by temperature, actinic flux spectrum, and concentrations of $O_3$, $HO_2$, and $RO_2$. However, since we do not have more detailed data, we are

not able to reconstruct the combined LCIE+EIE values at a higher time resolution in Freyer's work. Instead, we discussed this variation in the next section by using hourly NO, NO2 and j(NO2) data to model the daily variations.

What affect will the formation of PANs/addition of this equilibrium, have on the LCIE? Please comment on the LCIE that would be observed in the stratosphere.

This is a very good point. However, since 1) reactive nitrogen chemistry in the stratosphere is dramatically different from the troposphere and 2) the temperature in the stratosphere is also different, we suggest our model may not be used in the stratosphere. Therefore, we revised our statement and limited our discussion in the troposphere. In the meantime, we pointed out that future study is needed to explore the behavior of N isotopes in the stratosphere.

In the authors' experiment, NO is converted to NO2 by O3 in conditions with low concentrations of HO2 and RO2, which will play a role in the atmosphere. They suggest that the HO2 and RO2 oxidations of NO might have a similar KIE as the O3 oxidation, but this argument could be considered convenient. It would be stronger with experimental evidence and with improved validation by field measurements. Please make sure to discuss the potential uncertainty that is being introduced in transferring the laboratory results to the field.

We addressed this uncertainty by mentioning that, to enable our model in the ambient environment, we assume the KIE of O3 is similar to that of HO2 and RO2, because our modelled KIE value could explain the only field observation data by Freyer. This is an assumption that needs to be verified in the future.

However, if this assumption is true, then we do not need to know the concentrations of O3, HO2 or RO2, if we know both NO and NO2 concentrations. The existence of O3, HO2 or RO2 would be reflected in the NO/NO2 ratio at a given j(NO2) value.

No description is made of the UV lights that were used for the photolysis, please add

this. According to PHIFE/ZPE, photolytic isotopic fractionation changes as a function of wavelength. How did the spectrum of the lamps used to differ from the solar actinic flux spectrum? What wavelength dependence do you expect? How will the LCIE change as a function of altitude in the atmosphere as the actinic flux spectrum changes?

We added the description of the UV light in the supplementary material. And as mentioned before, we studied the variation of LCIE by calculating the PHIFE using 4 different actinic flux spectrums (lab light, early morning/late afternoon, mid-morning/afternoon, and noon). The results show that all of them have similar PHIFE values ranging from 1.0023 to 1.0029, suggesting the PHIFE do not vary significantly under different actinic flux spectrums.

There is precious little field data to use to validate the model. Please comment on what studies you would like to see in order to test the model, and as I have noted, please discuss the impact of different environmental factors, other than NOx concentration, on the results.

We added a paragraph before the final conclusion discussing the limitations of our work and potential future work that can be done to advance our understanding in this topic.

Presentation:

The abstract is rather short given the interesting findings of the paper. Please expand.

Revised as suggested. We rewrote the abstract to make it readable for a more general audience.

I am not sure why the TLA (three letter acronym) 'EIE' for 'equilibrium isotope effect' is introduced when there is already the widely accepted idea of the exchange reaction. This could make the abstract obscure for non-specialists.

Revised as suggested. We removed the TLA for EIE in the abstract but remained using EIE in the main text to contrast with LCIE.

[Figure]

The introduction should include discussion of photolytic re-emission of deposited nitrate.

Revised as suggested.

Please italicize the symbols used for physical quantities such as f, j and k.

Revised as suggested.

Please add a scheme or figure giving an overview of the key reactions involved in this work.

We added a sketch as Fig. 1C.

On page 11 line 230, some j values (photolysis rates) are presented. Please include the units with these numbers. Also, in figure 2, 'j = 0.005', but what are the units?

Revised as suggested.

Please also note the supplement to this comment:
https://www.atmos-chem-phys-discuss.net/acp-2019-1126/acp-2019-1126-AC1-supplement.pdf

---

## Author Comment (AC2) · 2 Apr 2020

Dear Dr. Savarino,

Thank you for your comment! This is an excellent suggestion that will improve our paper.

We have removed the discussion that used our model to explain the Arctic snow nitrate isotopes. Instead, we pointed out some uncertainties that still exist in this field, and we suggest that future work is needed to further address these issues.

[Figure]

2020.

---

## Author Comment (AC3) · 2 Apr 2020

Dear reviewer:

Thank you for the comments, we appreciate your input. We have revised our manuscript according to the comments from all the reviewers, below is our line-by-line response to your comments and suggestions. We also attached a PDF version of this response in case there is formatting issue.

First and foremost, how was the d15N of NOx measured (lines 150-151)? This is critical in that these values are used to demonstrate and calculate the observed isotope

effects.

We improved our description of measuring the 15N of NOx in the method section. In short, we measured the 15N of NOx in three different experiments. In each experiment, we inject same amount of NO and O3 to produce pure NO2, then we analyze the 15N value of the NO2. Because we can see that 100% of NOx in these experiments were in the form of NO2, therefore the measured 15N value can be used to represent the 15N of source NOx.

Second, was the d15N data of all samples corrected for potential isobaric in- fluences of 17O? (lines 120-124). The generated O3 should have a high D17O that will be transferred to the product NO2. This may impact both the starting NO source d15N values and the measured NO2 values during both dark and photochemical experiments. This could cause an important change in the findings if excess 17O has not been accounted for in correcting the d15N data. Related to the above, in Figure 1A it appears that the data was forced through an intercept of 0. But the best fit to the data does not appear to go through 0. What is the slope of the data not forced through the intercept? What might the intercept indicate – from my read this could indicate a shift due to the influence of 17O on the 45 signals when quantifying the isotopic ratios from N2O. A change in slope with the best fit might also bring the calculated value to something that is actually closer to the measured value by Walters et al. Wouldn't this be more consistent with current thinking?

The isobaric influence of O17 was calculated in the ISODAT system. It measures the 46 signals to calculate 18O first, then calculate the 17O value assuming mass dependent fractionation, then use these to correct for 15N signal. We notice that this method did not account for mass independent fractionation so it could shift the absolute 15N value (assuming O17 excess=30‰ by as much as 1.5‰Ḣowever, all the data shown on Figure 1A are (NO2)-(NOx), and both (NO2) and (NOx) have the same isobaric shift because they were measured using the same sampling & analysis method. So, when calculating the (NO2)-(NOx) values, this isobaric
error should be cancelled out assuming O17 excess were the same. Therefore, the isobaric error may not cause a significant shift in the interception on Figure 1A. In other words, our experiments that determined the 15N values of source NOx (in these experiments f(NO)=0 and (NO2)-(NOx)=0) can be seen as 3 extra data points at (0,0), therefore the interception on Figure 1A should still be 0.

For Figure 1B, the LCIE factor is calculated from "the best fit" (line 206). However, the figure makes it appear qualitative rather than quantitative. This calculation/estimation should be shown quantitatively and an r or r2 value should be reported for the fit. It should also be better explained why the point at AâĹij0.1 and AâĹij0.15 do not follow the expected relationship (why does the difference in d15N not change with A?). Also, why does the relationship have to be linear? In addition, the -10 per mil line does not "best fit" the field observation. It is also not clear what the error bars are on each symbol – I don't see how these should be the same/have the same meaning for the field observation versus the chamber data. The field study point should also be clearly cited in the figure caption. Related to this, in the text (line 215-218) it is argued that the experimental values determined are in good agreement with the field study. But the field study represents a very high NOx environment (at times NOx Âż O3) and the measurements were taken at different times of the year not just at temperature close to room temperature. So, is it possible that the field determined value is showing a temperature dependence relative to the controlled experiments?

We call the -10‰ LCIE line "best fit" because this fit gives the highest r value of 0.52 and the lowest total variation: total variation=$\sum (y\_i - f\_i)$ $2 in which y i is the observed value and f i is the predicted value by the fit line. We attribute the deviation of the two points from the$

Next, how does the EIE measurement compare with theoretical works? What may drive the differences in the experimental values versus theory? Is formation of other products (such as N2O3 or N2O4) in the chamber a concern and could that influence the measured EIE value?

We added some calculations showing the formation of N2O4 and N2O3 were negligible. Also, we have mentioned that we conducted a control experiment to evaluate NO2 wall loss but did not observe any NO2 loss over a 4-hour period. Therefore, we suggest the formation of other products were insignificant. We are not sure why it did not align with current theoretical calculations, probably because of the different approximation methods in previous studies. Hopefully future theoretical calculations can be carried out to evaluate our conclusions.

On line 185-189, it is stated that a2 ("alpha 2") is not quantified. However, Walters and Michalski (2016), which is referenced here, does include an ab initio-based estimate for this value. And couldn't this value be used to separate out the magnitude of PHIFE (rather than assume it)?

We added two paragraphs comparing our results to theoretical calculations. We pointed out that our result of -10‰ showed good agreement with theoretical calculations in Walters and Michalski (2016) and a ZPE approach that estimates the isotopic fractionation of NO2 photolysis.

Specific comments: Abstract – The abstract should be written for a more general audience. Be consistent with abbreviations. For example, NO and NO2 are not defined and nitrate isn't abbreviated. Also, LCIE should be more clearly defined since this is new terminology in this work. The implications of the study should be better highlighted here. How will a mathematic solution for NO-NO2 isotopic fractionation be useful to the atmospheric chemistry community?

We have revised our abstract, defined NO, NO2 and introduced LCIE. We also revised our implication and pointed out the limitation of this study.

Also, the statement that the new solution can be used at any given condition is a stretch since experiments were only conducted at room temperature and the experiments seem to be most relevant for the troposphere and not stratospheric NOx conditions.

We realized our experimentally determined values have limitations; therefore, we have discussed these limitations, and suggested our result should be applied in troposphere near room temperature. We also discussed how can future work verify and improve our current results.

Main Text

Line 25-30: The link between NOx and the formation of nitric acid (i.e. nitrate) needs to be more clearly stated. Also, the second sentence is a bit awkwardly phrased given that most of the studies did not use NO2 isotopes directly. It may be worth separating out the studies that have used isotopes of nitrate to understand something about NOx versus studies that have looked at NO2 or NOx specifically.

We revised the first paragraph. We now start this introduction by stating that the N isotopes are usually applied to study the sources of nitrate, however, it is unclear how atmospheric chemistry alters the isotope signals. Then, we narrow down our topic to the isotopic fractionations between NO and NO2, because it is very important.

Line 38: remove "the" before chemistry

Revised as recommended.

Lines 46-54: Please separate this into at least two separate sentences.

Revised as recommended.

Lines 61-63: What drives the difference in the theoretical predictions for this EIE?

It is mainly because each theoretical prediction uses different harmonic approximations in their calculation.

Lines 63-64: I think it should be pointed out that this was conducted at room temperatures. Also, the error of $\pm 0.001$ is incorrect?

Revised as recommended.

Lines 68-69: KIE and PHIFE for the NOx system is limited but you should probably acknowledge the KIE study on NO + O3: Walters and Michalski (2016) Ab initio study of nitrogen and position-specific oxygen kinetic isotope effects in the NO + O3 reaction, J. Chem. Phys. 145, 224307.

Revised as recommended.

Lines 70-75: suggest changing this to ". . .tends to diminish the expression of the equilibrium isotopic fractionation (EIE) between NO and NO2, but both KIE and PHIFE factors at that time were unknown." It seems strange to suggest that assuming no isotope effects (ie 1) yields no isotope effect. Here would be a good place to better detail the Freyer et al work and approach to determining the single fractionation factor. Otherwise line 75 doesn't really make sense to the reader unfamiliar with the details of Freyer's work.

We have revised this part to present a better description of Freyer's work and pointed out the limitation, which is the motivation of our study.

Line 83: atmospheric should be "atmospherically"

Revised as recommended.

Line 88: change "NOx nitrogen isotopes" to something more correct like isotopic composition of NOx.

Revised as recommended.

Line 105: "was" should be were. Line 105 (and later): what range of wavelengths are used in the experiment. This is important to report as you have already stated!

Revised as recommended.

Lines 112-115: More details on the capture of NO using denuder tubes should be provided in the text rather than supplement. And the details are not really given in the suppl either. What was the denuder coated with? How is it determined that there was

quantitative and accurate collection of the NO isotopes? What is this method based upon?

Revised as recommended.

Lines 120-124: Were corrections conducted for D17O interferences? I imagine the generated O3 will have a high D17O that will be proportional transferred to your product NO2. This may impact both your starting NO source d15N value and your measured NO2 values during dark and photochemical experiments. Section 2 overall – were any blanks tested throughout the experiments?

The D17O will affect the measured absolute ′15N values but this should be cancelled out when we calculate the ′(NO2)-′(NOx) values (see our reply above). We tested 6 blanks during our experiments and none of them showing any measurable nitrite. We have added this part into the main text.

Line 129: difference should be "ratio of" correct?

Revised as recommended.

Line 140: I don't understand the formatting here with d(15N, NO)?

We changed this notation to ′15N(NO).

Line 149- 150: How was d15N-NOx measured? If this was done for all experiments, why does n=3? and again how will potential 17O isobaric influences affect your quantification of d15N?

As we described above, three extra experiments have been conducted in which we convert all the NO into NO2 and measured the ′15N values of NO2 to represent the ′15N of NOx. All three experiments showed consistent ′15N values, therefore we suggest the ′15N value of NOx remain the same in all of our experiments.

Lines 156: Where does the error on the 26.8 value come from? This is not represented in the figure.

Since the slope actually represents the ˊ(NO2)-ˊ(NO) values in each experiment, we calculated the error using the standard deviations of ˊ(NO2)-ˊ(NO) values in the 5 experiments.

Lines 166-167: Can you prove that formation of these other products are not important for the experiments performed here? EIE should be able to be precisely predicted by theory – so why is there such a mismatch between the theoretical and the measured values?

We added some calculations showing the formation of N2O4 and N2O3 were negligible. Also, we have mentioned that we conducted a control experiment to evaluate NO2 wall loss but did not observe any NO2 loss over a 4-hour period. Therefore, we suggest the formation of other products were insignificant. We are not sure why it did not align with current theoretical calculations, probably because of the different approximation methods in previous studies. We suggest that future theoretical calculations can be carried out to evaluate our conclusions.

Figure 1A: It appears that the data was fitted through an intercept of 0, but the data doesn't appear to go through the intercept. What is the slope of the data when not forced through the intercept? What might the intercept of your data indicate? Maybe D17O isobaric influence?

Since the ˊ(NOx) is determined using the same method as ˊ(NO2) in our experiments, this isobaric influence in this equation should be cancelled out (see our reply above). Therefore, this slope should have an intercept of 0.

Lines 174-176: What exactly were the wavelengths of the blacklight used in the experiments?

We provided an irradiation spectrum of the UV light we used in the supplementary material.

Lines 186-189: $\alpha 2$ value was determined in Walters and Michalski, 2016 ab initio study

as referenced above.

We changed the statement to "nor were $\alpha1$ and $\alpha2$ experimentally determined"

Lines 191-197: It might be more straightforward if t(exchange) and t(photo) were defined first and then A, etc. Also please better define the purpose of equations 7 and 8. Also shouldn't these equations have epsilon instead of alpha?

Revised as recommended. In these calculations, since =($\alpha$-1)*1000‰ we know 2-1=($\alpha$2-$\alpha$1)*1000‰ To introduce as little symbols as possible, we did not use  in this section.

Line 207-210: The experimental LCIE should be compared with the NO + O3 KIE. Here and on line 225 it feels a bit like the large uncertainty of +/- 50 percent on the -10 value is being ignored!

We added some extra discussion suggesting 1) this -10‰ value fits well with theoretical predictions and 2) the $\alpha$1 value did not vary significantly with a changing j(NO2). Therefore, we will use this -10‰ in the following discussion assuming the $\alpha$1 value remain constant, and 2) the NO+RO2/HO2 reactions have the same fractionation factors ($\alpha$2) as NO+O3.

Line 220: Note that the field experiments sometimes only represent NO2 and other times NOx. . .so the difference between NO and NO2 was not measured, it was determined.

Revised as recommended.

Line 237: I do not see how this shown in Figure 2A.

Figure 2A represents the isotopic fractionations between NO and NO2 in dark condition (j(NO2)=0). In this scenario, EIE solely controls the isotopic fractionation therefore the (NO2)-(NO) should be a constant no matter how NOx level and f(NO2) changes.

Line 261-263: These are not emissions, these are ambient NO2 and NOx. Also, NOx

is not emitted – primary emissions are NO and very on occasion diesel engines have been shown to emit NO2 directly. All of the language here needs to be much more precise.

We changed "NOx emission" to "total NOx".

Section 4 overall: This section needs work. It needs to be established why the sites were chosen. And the sites should be clearly labeled as representative of, for instance, urban versus suburban versus remote versus elevated NOx (roadside) or the like. It seems like the point here actually should be to distinguish the effects across the different sites. Why does it not matter that the O3 (and HO2, RO2, etc) concentrations would vary significantly across these sites?

The four sites represented different NOx level and we can see the impact of NOx level to the NO-NO2 isotopic fractionations was significant. O3 (and HO2, RO2) concentrations impact the NO-NO2 fractionation by altering the A values, which was reflected in the f(NO) parameter and the A value in our equations. From Eq. 7 and 8 we can know that we do not need to use O3 (or HO2, RO2) concentration to calculate the isotopic fractionations as long as we know NO and NO2 concentrations and the j(NO2) values.

What are we really learning from site A vs B? They are both roadside. Given that there is such limited data to compare the model with, could the authors compare their roadside model to d15N(NO2) data collected by a roadside such as Felix and Elliott, 2014, "Isotopic composition of passively collected nitrogen dioxide emissions: Vehicle, soil, and livestock source signatures", Atmospheric Environment, 92, 359-366?

A and B are both roadside sites, however they have different NOx concentrations. The NOx concentrations at Anaheim site averaged at 58 ppb but the Evansville site only had 15 ppb. As a result, the isotopic fractionations at Anaheim was mainly controlled by EIE which showed high (NO2)-(NOx) values (>10‰ throughout the day), but at Evansville, LCIE was more significant, thus the (NO2)- (NOx) values can be as low as ∼2‰ at noon. Our discussion was less focused on the 15N values of

[Figure]

NOx sources but more focused on the impact of the NOx level and photochemistry to the isotopic fractionations between NO and NO2. Felix and Elliott, 2014 provided a good insight on the 15N(NO2) values at roadside, and the NOx level at Felix and Elliott, 2014 study was similar to that of Anaheim. Therefore, we suggest at these sites, EIE will also be a dominate factor. Furthermore, in Felix and Elliott, 2014, there are little constrain on the differences between the 15N values of NO2 and total NOx, thus we are not able to further compare our work to theirs.

Also why not directly compare with what the model would predict quantitatively against the Walters 2018 work? And also predict the values for the Freyer work and compare?

Both Walters et al. 2018 and Freyer 1993 work used the same equation to calculate the isotopic shift which is similar to our approach. The only difference between their equation and our equation is that they assumed $\alpha2\text{-}\alpha1=0$ instead of the -10‰ measured in our experiment. We can see in these two works, the calculated isotopic shift values are similar to our results, suggesting the differences in LCIE may only slightly impact our results by a couple per mil in these conditions.

Lines 325-327: This conclusion is a bit strange. There is very little local HNO3 at Summit, Greenland. So drawing the conclusion based upon snow work (not atmosphere and snow) and assuming a direct link temporally between d15NO2 and d15NO3- seems a stretch. It might be more useful to look at Jarvis et al. (2009) instead – Jarvis, J. C., M. G. Hastings, E. J. Steig, and S. A. Kunasek (2009), Isotopic ratios in gas- phase HNO3 and snow nitrate at Summit, Greenland, J. Geophys. Res., 114, D17301, doi:10.1029/2009JD012134.

This is very good point. I have removed this part because directly using our model to interpret these data is not solid enough.

Line 329: what kinds of data and future environmental measurements should be conducted to validate this model? Help the community make this a reality!

We added a paragraph in the end of this section to address our limitations and future work. We suggest that future experiments, field observations and theoretical studies should be done to 1) verify our experimental results, 2) investigate the isotopic fractionation factor of reactions that converts NOx into NOy and nitrate, 3) study the isotopic effects of reactive nitrogen chemistry in the stratosphere amd 4) study the temperature dependence of these fractionation factors.

Supplement – This needs to be re-read and edited – there are a lot of typos. Some comments on the methods in the supplement – I have a number of questions: -what was the flow rate used to calculate the NO2 residence time ("reside" should be residence in the text)? -include more details of the denuder method – what company are these from? What were they coated with? How do you know they are quantitative in collection? Please show the collection efficiency data. And report how many times this was tested. -need to make it clearer where the 1.0268 value comes from – why is this value assumed here? Based upon the "best fit"? which really isn't a best fit (see comments from above). So what happens here if you do not assume a forced zero intercept? -make sure the editor's suggested technical corrections carry through the supplement as well.

We have gone through the supplementary material and fixed some typos. To answer the questions: we provided more detailed descriptions (e.g., flow rate, information about the denuder method) both in the main text and in the supplementary material. In the meantime, we have shown that our collection efficiency was nearly 100% by stating that we measured the NO2 level on the exit of the denuder tubes when using the denuder tubes to collect NO2 at 66 ppb, and the measured NO2 level at the denuder exit was below detection limit. We only conducted this control experiment once, but the testing lasts for over an hour, and the collection efficiency remained at 100% throughout this experiment. We also changed the 1.0268 to 1.0275 to align with the main text, and the reasons we used a zero intercept have been stated above.

Please also note the supplement to this comment:
https://www.atmos-chem-phys-discuss.net/acp-2019-1126/acp-2019-1126-AC3-supplement.pdf

**Supplement:**

**Response to RC2**

Dear reviewer:

      Thank you for the comments, we appreciate your input. We have revised our manuscript according to the comments from all the reviewers, below is our line-by-line response to your comments and suggestions.

First and foremost, how was the d15N of NOx measured (lines 150-151)? This is critical in that these values are used to demonstrate and calculate the observed isotope effects.

      We improved our description of measuring the $\delta^{15}N$ of $NO_x$ in the method section. In short, we measured the $\delta^{15}N$ of $NO_x$ in three different experiments. In each experiment, we inject same amount of NO and $O_3$ to produce pure $NO_2$, then we analyze the $\delta^{15}N$ value of the $NO_2$. Because we can see that 100% of $NO_x$ in these experiments were in the form of $NO_2$, therefore the measured $\delta^{15}N$ value can be used to represent the $\delta^{15}N$ of source $NO_x$.

Second, was the d15N data of all samples corrected for potential isobaric in- fluences of 17O? (lines 120-124). The generated O3 should have a high D17O that will be transferred to the product NO2. This may impact both the starting NO source d15N values and the measured NO2 values during both dark and photochemical experiments. This could cause an important change in the findings if excess 17O has not been accounted for in correcting the d15N data. Related to the above, in Figure 1A it appears that the data was forced through an intercept of 0. But the best fit to the data does not appear to go through 0. What is the slope of the data not forced through the intercept? What might the intercept indicate – from my read this could indicate a shift due to the influence of 17O on the 45 signals when quantifying the isotopic ratios from N2O. A change in slope with the best fit might also bring the calculated value to something that is actually closer to the measured value by Walters et al. Wouldn't this be more consistent with current thinking?

      The isobaric influence of O17 was calculated in the ISODAT system. It measures the 46 signals to calculate $\delta^{18}O$ first, then calculate the $\delta^{17}O$ value assuming mass dependent fractionation, then use these to correct for $\delta^{15}N$ signal. We notice that this method did not account for mass independent fractionation so it could shift the absolute $\delta^{15}N$ value (assuming O17 excess=30‰) by as much as 1.5‰.

      However, all the data shown on Figure 1A are $\delta(NO_2)-\delta(NO_x)$, and both $\delta(NO_2)$ and $\delta(NO_x)$ have the same isobaric shift because they were measured using the same sampling & analysis method. So, when calculating the $\delta(NO_2)-\delta(NO_x)$ values, this isobaric error should be cancelled out assuming O17 excess were the same. Therefore, the isobaric error may not cause a significant shift in the interception on Figure 1A.

      In other words, our experiments that determined the $\delta^{15}N$ values of source $NO_x$ (in these experiments f(NO)=0 and $\delta(NO_2)-\delta(NO_x)=0$) can be seen as 3 extra data points at (0,0), therefore the interception on Figure 1A should still be 0.

For Figure 1B, the LCIE factor is calculated from "the best fit" (line 206). However, the figure makes it appear qualitative rather than quantitative. This calculation/estimation should be shown quantitatively and an r or r2 value should be reported for the fit. It should also be better explained

why the point at A~0.1 and A~0.15 do not follow the expected relationship (why does the difference in d15N not change with A?). Also, why does the relationship have to be linear? In addition, the -10 per mil line does not "best fit" the field observation. It is also not clear what the error bars are on each symbol – I don't see how these should be the same/have the same meaning for the field observation versus the chamber data. The field study point should also be clearly cited in the figure caption. Related to this, in the text (line 215-218) it is argued that the experimental values determined are in good agreement with the field study. But the field study represents a very high NOx environment (at times NOx » O3) and the measurements were taken at different times of the year not just at temperature close to room temperature. So, is it possible that the field determined value is showing a temperature dependence relative to the controlled experiments?

We call the -10‰ LCIE line "best fit" because this fit gives the highest r value of 0.52 and the lowest total variation:

total variation=$\sum(y_i - f_i)^2$

in which $y_i$ is the observed value and $f_i$ is the predicted value by the fit line.

We attribute the deviation of the two points from the prediction line to the relatively large analytical uncertainties at low A values. In these two experiments, the NO and $NO_2$ level were low (<10 ppb), and the concentration measurements showed a higher error bar. We have recalculated the error bars on these data points which are now shown on Figure 1B.

We used the average conditions of the field study to calculate its position at Figure 2B. Although the conditions change significantly throughout the time period of this study, their 18.8‰ value represented the average fractionation factor of the sampling period. Therefore, we also used their average condition to calculate the fractionation factor. However, it is likely that the temperature dependence played a role in this study, and we pointed out that future studies are needed to investigate its impact.

Next, how does the EIE measurement compare with theoretical works? What may drive the differences in the experimental values versus theory? Is formation of other products (such as N2O3 or N2O4) in the chamber a concern and could that influence the measured EIE value?

We added some calculations showing the formation of $N_2O_4$ and $N_2O_3$ were negligible. Also, we have mentioned that we conducted a control experiment to evaluate $NO_2$ wall loss but did not observe any $NO_2$ loss over a 4-hour period. Therefore, we suggest the formation of other products were insignificant.

We are not sure why it did not align with current theoretical calculations, probably because of the different approximation methods in previous studies. Hopefully future theoretical calculations can be carried out to evaluate our conclusions.

On line 185-189, it is stated that a2 ("alpha 2") is not quantified. However, Walters and Michalski (2016), which is referenced here, does include an ab initio-based estimate for this value. And couldn't this value be used to separate out the magnitude of PHIFE (rather than assume it)?

We added two paragraphs comparing our results to theoretical calculations. We pointed out that our result of -10‰ showed good agreement with theoretical calculations in Walters and Michalski (2016) and a ZPE approach that estimates the isotopic fractionation of $NO_2$ photolysis.

Specific comments: Abstract – The abstract should be written for a more general audience. Be consistent with abbreviations. For example, NO and NO2 are not defined and nitrate isn't abbreviated. Also, LCIE should be more clearly defined since this is new terminology in this work. The implications of the study should be better highlighted here. How will a mathematic solution for NO-NO2 isotopic fractionation be useful to the atmospheric chemistry community?

We have revised our abstract, defined NO, $NO_2$ and introduced LCIE. We also revised our implication and pointed out the limitation of this study.

Also, the statement that the new solution can be used at any given condition is a stretch since experiments were only conducted at room temperature and the experiments seem to be most relevant for the troposphere and not stratospheric NOx conditions.

We realized our experimentally determined values have limitations; therefore, we have discussed these limitations, and suggested our result should be applied in troposphere near room temperature. We also discussed how can future work verify and improve our current results.

Main Text

Line 25-30: The link between NOx and the formation of nitric acid (i.e. nitrate) needs to be more clearly stated. Also, the second sentence is a bit awkwardly phrased given that most of the studies did not use NO2 isotopes directly. It may be worth separating out the studies that have used isotopes of nitrate to understand something about NOx versus studies that have looked at NO2 or NOx specifically.

We revised the first paragraph. We now start this introduction by stating that the N isotopes are usually applied to study the sources of nitrate, however, it is unclear how atmospheric chemistry alters the isotope signals. Then, we narrow down our topic to the isotopic fractionations between NO and $NO_2$, because it is very important.

Line 38: remove "the" before chemistry

Revised as recommended.

Lines 46-54: Please separate this into at least two separate sentences.

Revised as recommended.

Lines 61-63: What drives the difference in the theoretical predictions for this EIE?

It is mainly because each theoretical prediction uses different harmonic approximations in their calculation.

Lines 63-64: I think it should be pointed out that this was conducted at room temperatures. Also, the error of ±0.001 is incorrect?

Revised as recommended.

Lines 68-69: KIE and PHIFE for the NOx system is limited but you should probably acknowledge the KIE study on NO + O3: Walters and Michalski (2016) Ab initio study of nitrogen and position-specific oxygen kinetic isotope effects in the NO + O3 reaction, J. Chem. Phys. 145, 224307.

Revised as recommended.

Lines 70-75: suggest changing this to ". . .tends to diminish the expression of the equilibrium isotopic fractionation (EIE) between NO and NO2, but both KIE and PHIFE factors at that time were unknown." It seems strange to suggest that assuming no isotope effects (ie 1) yields no isotope effect. Here would be a good place to better detail the Freyer et al work and approach to determining the single fractionation factor. Otherwise line 75 doesn't really make sense to the reader unfamiliar with the details of Freyer's work.

We have revised this part to present a better description of Freyer's work and pointed out the limitation, which is the motivation of our study.

Line 83: atmospheric should be "atmospherically"

Revised as recommended.

Line 88: change "NOx nitrogen isotopes" to something more correct like isotopic composition of NOx.

Revised as recommended.

Line 105: "was" should be were. Line 105 (and later): what range of wavelengths are used in the experiment. This is important to report as you have already stated!

Revised as recommended.

Lines 112-115: More details on the capture of NO using denuder tubes should be provided in the text rather than supplement. And the details are not really given in the suppl either. What was the denuder coated with? How is it determined that there was quantitative and accurate collection of the NO isotopes? What is this method based upon?

Revised as recommended.

Lines 120-124: Were corrections conducted for D17O interferences? I imagine the generated O3 will have a high D17O that will be proportional transferred to your product NO2. This may impact both your starting NO source d15N value and your measured NO2 values during dark and photochemical experiments. Section 2 overall – were any blanks tested throughout the experiments?

The D17O will affect the measured absolute $\delta^{15}N$ values but this should be cancelled out when we calculate the $\delta(NO_2)-\delta(NO_x)$ values (see our reply above). We tested 6 blanks during our

experiments and none of them showing any measurable nitrite. We have added this part into the main text.

Line 129: difference should be "ratio of" correct?

Revised as recommended.

Line 140: I don't understand the formatting here with d(15N, NO)?

We changed this notation to $\delta^{15}N(NO)$.

Line 149- 150: How was d15N-NOx measured? If this was done for all experiments, why does n=3? and again how will potential 17O isobaric influences affect your quantification of d15N?

As we described above, three extra experiments have been conducted in which we convert all the NO into $NO_2$ and measured the $\delta^{15}N$ values of $NO_2$ to represent the $\delta^{15}N$ of $NO_x$. All three experiments showed consistent $\delta^{15}N$ values, therefore we suggest the $\delta^{15}N$ value of $NO_x$ remain the same in all of our experiments.

Lines 156: Where does the error on the 26.8 value come from? This is not represented in the figure.

Since the slope actually represents the $\delta(NO_2)-\delta(NO)$ values in each experiment, we calculated the error using the standard deviations of $\delta(NO_2)-\delta(NO)$ values in the 5 experiments.

Lines 166-167: Can you prove that formation of these other products are not important for the experiments performed here? EIE should be able to be precisely predicted by theory – so why is there such a mismatch between the theoretical and the measured values?

We added some calculations showing the formation of $N_2O_4$ and $N_2O_3$ were negligible. Also, we have mentioned that we conducted a control experiment to evaluate $NO_2$ wall loss but did not observe any $NO_2$ loss over a 4-hour period. Therefore, we suggest the formation of other products were insignificant.
We are not sure why it did not align with current theoretical calculations, probably because of the different approximation methods in previous studies. We suggest that future theoretical calculations can be carried out to evaluate our conclusions.

Figure 1A: It appears that the data was fitted through an intercept of 0, but the data doesn't appear to go through the intercept. What is the slope of the data when not forced through the intercept? What might the intercept of your data indicate? Maybe D17O isobaric influence?

Since the $\delta(NO_x)$ is determined using the same method as $\delta(NO_2)$ in our experiments, this isobaric influence in this equation should be cancelled out (see our reply above). Therefore, this slope should have an intercept of 0.

Lines 174-176: What exactly were the wavelengths of the blacklight used in the experiments?

We provided an irradiation spectrum of the UV light we used in the supplementary material.

Lines 186-189: $\alpha2$ value was determined in Walters and Michalski, 2016 ab initio study as referenced above.

We changed the statement to "nor were $\alpha1$ and $\alpha2$ experimentally determined"

Lines 191-197: It might be more straightforward if t(exchange) and t(photo) were defined first and then A, etc. Also please better define the purpose of equations 7 and 8. Also shouldn't these equations have epsilon instead of alpha?

Revised as recommended. In these calculations, since $\varepsilon=(\alpha-1)*1000‰$, we know $\varepsilon2-\varepsilon1=(\alpha2-\alpha1)*1000‰$. To introduce as little symbols as possible, we did not use $\varepsilon$ in this section.

Line 207-210: The experimental LCIE should be compared with the NO + O3 KIE. Here and on line 225 it feels a bit like the large uncertainty of +/- 50 percent on the -10 value is being ignored!

We added some extra discussion suggesting 1) this -10‰ value fits well with theoretical predictions and 2) the $\alpha_1$ value did not vary significantly with a changing $j(NO_2)$. Therefore, we will use this -10‰ in the following discussion assuming the $\alpha_1$ value remain constant, and 2) the $NO+RO_2/HO_2$ reactions have the same fractionation factors ($\alpha_2$) as $NO+O_3$.

Line 220: Note that the field experiments sometimes only represent NO2 and other times NOx. . .so the difference between NO and NO2 was not measured, it was determined.

Revised as recommended.

Line 237: I do not see how this shown in Figure 2A.

Figure 2A represents the isotopic fractionations between NO and $NO_2$ in dark condition ($j(NO_2)=0$). In this scenario, EIE solely controls the isotopic fractionation therefore the $\delta(NO_2)-\delta(NO)$ should be a constant no matter how $NO_x$ level and f(NO2) changes.

Line 261-263: These are not emissions, these are ambient NO2 and NOx. Also, NOx is not emitted – primary emissions are NO and very on occasion diesel engines have been shown to emit NO2 directly. All of the language here needs to be much more precise.

We changed "$NO_x$ emission" to "total $NO_x$".

Section 4 overall: This section needs work. It needs to be established why the sites were chosen. And the sites should be clearly labeled as representative of, for instance, urban versus suburban versus remote versus elevated NOx (roadside) or the like. It seems like the point here actually should be to distinguish the effects across the different sites. Why does it not matter that the O3 (and HO2, RO2, etc) concentrations would vary significantly across these sites?

The four sites represented different $NO_x$ level and we can see the impact of NOx level to the $NO$-$NO_2$ isotopic fractionations was significant.

$O_3$ (and $HO_2$, $RO_2$) concentrations impact the $NO$-$NO_2$ fractionation by altering the A values, which was reflected in the f(NO) parameter and the A value in our equations. From Eq. 7 and 8 we can know that we do not need to use $O_3$ (or $HO_2$, $RO_2$) concentration to calculate the isotopic fractionations as long as we know NO and $NO_2$ concentrations and the $j(NO_2)$ values.

What are we really learning from site A vs B? They are both roadside. Given that there is such limited data to compare the model with, could the authors compare their roadside model to d15N(NO2) data collected by a roadside such as Felix and Elliott, 2014, "Isotopic composition of passively collected nitrogen dioxide emissions: Vehicle, soil, and livestock source signatures", Atmospheric Environment, 92, 359-366?

A and B are both roadside sites, however they have different $NO_x$ concentrations. The $NO_x$ concentrations at Anaheim site averaged at 58 ppb but the Evansville site only had 15 ppb. As a result, the isotopic fractionations at Anaheim was mainly controlled by EIE which showed high $\delta(NO_2)$-$\delta(NO_x)$ values (>10‰ throughout the day), but at Evansville, LCIE was more significant, thus the $\delta(NO_2)$- $\delta(NO_x)$ values can be as low as ~2‰ at noon.

Our discussion was less focused on the $\delta^{15}N$ values of $NO_x$ sources but more focused on the impact of the $NO_x$ level and photochemistry to the isotopic fractionations between NO and $NO_2$. Felix and Elliott, 2014 provided a good insight on the $\delta^{15}N(NO_2)$ values at roadside, and the NOx level at Felix and Elliott, 2014 study was similar to that of Anaheim. Therefore, we suggest at these sites, EIE will also be a dominate factor. Furthermore, in Felix and Elliott, 2014, there are little constrain on the differences between the $\delta^{15}N$ values of $NO_2$ and total $NO_x$, thus we are not able to further compare our work to theirs.

Also why not directly compare with what the model would predict quantitatively against the Walters 2018 work? And also predict the values for the Freyer work and compare?

Both Walters et al. 2018 and Freyer 1993 work used the same equation to calculate the isotopic shift which is similar to our approach. The only difference between their equation and our equation is that they assumed $\alpha2$-$\alpha1$=0 instead of the -10‰ measured in our experiment. We can see in these two works, the calculated isotopic shift values are similar to our results, suggesting the differences in LCIE may only slightly impact our results by a couple per mil in these conditions.

Lines 325-327: This conclusion is a bit strange. There is very little local HNO3 at Summit, Greenland. So drawing the conclusion based upon snow work (not atmosphere and snow) and assuming a direct link temporally between d15NO2 and d15NO3- seems a stretch. It might be more useful to look at Jarvis et al. (2009) instead – Jarvis, J. C., M. G. Hastings, E. J. Steig, and S. A. Kunasek (2009), Isotopic ratios in gas- phase HNO3 and snow nitrate at Summit, Greenland, J. Geophys. Res., 114, D17301, doi:10.1029/2009JD012134.

This is very good point. I have removed this part because directly using our model to interpret these data is not solid enough.

Line 329: what kinds of data and future environmental measurements should be conducted to validate this model? Help the community make this a reality!

We added a paragraph in the end of this section to address our limitations and future work. We suggest that future experiments, field observations and theoretical studies should be done to 1) verify our experimental results, 2) investigate the isotopic fractionation factor of reactions that converts $NO_x$ into $NO_y$ and nitrate, 3) study the isotopic effects of reactive nitrogen chemistry in the stratosphere amd 4) study the temperature dependence of these fractionation factors.

Supplement – This needs to be re-read and edited – there are a lot of typos. Some comments on the methods in the supplement – I have a number of questions: -what was the flow rate used to calculate the NO2 residence time ("reside" should be residence in the text)? -include more details of the denuder method – what company are these from? What were they coated with? How do you know they are quantitative in collection? Please show the collection efficiency data. And report how many times this was tested. -need to make it clearer where the 1.0268 value comes from – why is this value assumed here? Based upon the "best fit"? which really isn't a best fit (see comments from above). So what happens here if you do not assume a forced zero intercept? -make sure the editor's suggested technical corrections carry through the supplement as well.

We have gone through the supplementary material and fixed some typos. To answer the questions: we provided more detailed descriptions (e.g., flow rate, information about the denuder method) both in the main text and in the supplementary material. In the meantime, we have shown that our collection efficiency was nearly 100% by stating that we measured the $NO_2$ level on the exit of the denuder tubes when using the denuder tubes to collect $NO_2$ at 66 ppb, and the measured NO2 level at the denuder exit was below detection limit. We only conducted this control experiment once, but the testing lasts for over an hour, and the collection efficiency remained at 100% throughout this experiment. We also changed the 1.0268 to 1.0275 to align with the main text, and the reasons we used a zero intercept have been stated above.

---

## Editor Decision (ED1)

Dear Dr Li

Many thanks for submitting your further revised manuscript, which I am happy to accept for publication subject to a few minor revisions as outlined below.

1) Please integrate the supplement into the main text as appendix or appendices. The ACP manuscript guidelines require that "Supplementary material is reserved for items that cannot reasonably be included in the main text or as appendices. These may include short videos, very large images, maps, CIF files, as well as short computer codes such as matlab or python script. [...] Normal size figures, tables, as well as technical or theoretical developments that do not need to be included in the main text should be included as appendices."

2) Please remove the unnecessary approximations made in deriving equations 7 and 8, and make them consistent with Eq. 3. I have included the final steps of the calculation without approximation here (using the same $\delta$ and $\varepsilon$ symbols as in the calculation I sent you during the discussion stage). Obviously, your steps up to the ratio $R/R_2$ are fine (except that I have used the inverse ratio, to simplify the final step of the calculation):

Mass balance:

$$f\delta(NO) + (1-f)\delta(NO_2) = \delta(NO_x)$$

$$f\delta + (1-f)(1+\delta_2) = \delta_x$$

$$\delta_2 - \delta_x = f(\delta_2 - \delta)$$

with $f = f(NO)$

Steady-state assumption to derive Eq. 7:

$$\frac{R}{R_2} - 1 = \frac{1+\delta}{1+\delta_2} - 1 = \frac{1+A\alpha_1}{1+A\alpha_2+\varepsilon} - 1$$

$$\frac{\delta - \delta_2}{1+\delta_2} = \frac{A(\alpha_1 - \alpha_2) - \varepsilon}{1+A\alpha_2+\varepsilon}$$

$$\delta_2 - \delta = (1+\delta_2)\frac{(\alpha_2 - \alpha_1)A + \varepsilon}{1+A\alpha_2+\varepsilon}$$

Steady-state assumption combined with mass-balance to derive Eq. 8:

$$\delta_2 - \delta_x = (1+\delta_2)f\frac{(\alpha_2 - \alpha_1)A + \varepsilon}{1+A\alpha_2+\varepsilon}$$

$$= (1+\delta_2)[1 - f(NO_2)]\frac{(\alpha_2 - \alpha_1)A + \varepsilon}{1+A\alpha_2+\varepsilon}$$

Setting $A = 0$ (very fast isotope exchange) immediately reproduces Eq. 3 for verification.

3)      There is no need to repeat the reaction equations (R1 to R6, p. 5 of current supplement) or the definition of delta values (p. 6 of current supplement, lines from "Next, to calculate ..." to "$R_{NO2}/R_{NO} - 1$") in the Appendix.

4)      Please remove unnecessary factors ("1000 ‰") from your mathematical equations (e.g. l. 180 and 182).

5)      Please also remove the unnecessary multiplication symbols (×). These make the equations unnecessarily difficult to read. Such multiplication symbols are rarely required outside scientific notation, e.g. $8 \times 10^{-14}$ $cm^3$ $s^{-1}$.

6)      L. 182: The quantity with the symbol $\varepsilon$ should not be called "isotope enrichment factor" (which is a different kind of quantity). Please use the term "equilibrium isotopic fractionation" instead (as you do on l. 89 and elsewhere).

7)      L. 188 & 193: The quantities should be enclosed in parentheses so that the unit applies to value and uncertainty, e.g. "$(-58.7\pm0.8)$ ‰"

Kind regards
Jan Kaiser
Editor ACP

---

## Author Response (AR2)

**Response to Referee 2:**

The manuscript's abstract, introduction and conclusions are much improved based upon responses to the reviewer comments. A few important issues remain.

It's a bit of a circular argument to suggest that because they can "fit" the Freyer et al 1993 observational value for the combined LCIE and EIE of ~ -18 per mil that this justifies their experimental value for the combined LCIE of -10 per mil. There are several ways in which the study needs to be more quantitative in its approach to evaluating the experimental results. Currently, overall, I am unimpressed with the application to environmental data because the data under heavily controlled experimental conditions shows a great deal of variability that is not discussed. How will we ever understand this in the field, with a wide variety of complications versus the chamber experiments if we cannot even fully address the uncertainties associated with the controlled experiments?

We agree with the reviewer's point that there are still uncertainties hindering us from fully model the $\delta^{15}N$ values of atmospheric reactive nitrogen. We would like to measure the EIE and LCIE at other temperatures in the future to advance our current understanding. However, we believe our equations that quantifying the relative importance of EIE and LCIE in the $NO-NO_2$ system should be valid although the constants could change by environment. There are a lot more work still needed to fill the knowledge gap and we are looking forward to future investigations.

Figure 1A and 1B should be dissected QUANTITATIVELY. Best fit terminology is used in the manuscript in a here qualitative manner. For Figure 1A, the actual best linear fit to the experimental results should be quantified and compared against the forced intercept line that is currently pictured. The standard error for both the slope and intercept should be calculated for the best linear fit to the data and this should be compared to assess whether the experimental results are statistically significantly different than (0,0). It appears it might be significantly different, and this should be discussed if it is (i.e., what causes the difference from (0,0)? how might this be better addressed in future experiments?).

We agree with the reviewer's point that we need to provide quantification to our fit lines in both EIE and LCIE experiments. To better demonstrate this, we added some description of the uncertainties in the figure caption and we pointed out the error bars on the figures represented the overall uncertainties of $NO_x$ concentration measurement and isotopic analysis. For the EIE line, we provided the standard error (1.2‰) and for the LCIE line we reported RMSE (1.1‰) and suggest the -10‰ line showed the lowest RMSE when comparing to our measured data.

The response to reviewers includes discussion of the error bars in Figures 1A and 1B, but this also needs to be discussed in the main text and should be indicated in the caption what the error bars represent.

Revised as suggested.

For Figure 1B, the -10 line (which is actually -10.5 in the figure) is simply not the best qualitative fit to both the experimental data and the Freyer observation. Even qualitatively the language should be changed to better address what you are trying to prove here. Better yet, I suggest quantitatively calculating what line actually fits the experimental data…you could even do this while including and excluding the Freyer datapoint. Then address how the actual best fit through the data compared with the theoretical (case-based) lines drawn in Figure 1B.

We agree with the reviewer's point that we need to provide quantification to our fit lines. In the LCIE figure, we used Rooted Mean Square Error (RMSE) to assess the fit and the -10‰ line gives the lowest RMSE of 1.1‰. We have added this part into the main text. Also, we have changed the number in the figure because with 5‰ uncertainty, the last digit has no meaning.

The lack of correction for isobaric influence needs to be directly addressed in the Methods section. This could in fact make a difference on the order of 0.5-0.8 per mil under the equations used here. While the non-zero D17O correction to d15N is probably not making a difference in the dark experiments, it could make a difference in the photochemical experiments. It is clear in the literature regarding measurement of d15N from N2O that this correction is important and there is no reason for it not at least be addressed here.

The isobaric correction when measuring atmospheric $NO_2$ is important, especially when $O_3$ participated in the reactions. In our experiment, $O_3$ played an important role during all the conversion between NO and $NO_2$ therefore we agree that all of our samples should somewhat show some isobaric error.

However, we suggest the isobaric effects should be cancelled out in our graphs when we subtract $\delta^{15}N$ value of measured $NO_2$ with $\delta^{15}N$ value of measured $NO_x$. When we measure the $\delta^{15}N$ of source NOx, we first injected O3 into the chamber then injected NO and ensure 100% of NO was converted into $NO_2$. During this process, the measured $\delta(NO_x)$ also carried the isobaric error. Thus, when we subtract $\delta(NO_2)$ with $\delta(NO_x)$, this error is likely cancelled out. This is also the reason that the five data points should be forced to have an intercept of 0 because measuring $\delta(NO_x)$ can be seen as three individual experiments that fall on (0,0) in the figure.

The addition of Figure 1C is nice, but I would suggest adding it as Figure 1A and referring to it much earlier on (when defining EIE, LCIE and PHIFE) and then referring to it again after quantifying the values and noting that they are on the figure – the figure caption could also include the values in the figure are determined "in this work".

Revised as suggested.

Lastly, I am confused by mention (in the response to reviewers) of Walters and Michalski assuming alpha1-alpha2 = 0. My understanding is that the alpha1 is assumed to be 1 such that the fractionation factor is 0; alpha2 is theoretically determined with an epsilon ~ -7 per mil. So that would suggest that a1-a2 does not = 0. Please revisit this to be sure that this misinterpretation is not confusing the results/discussion in the manuscript.

Sorry for the confusion. We have pointed out the alpha 2 values of -7‰ in the main text and suggested a good match between the experimentally determined value (LCIE=-10±5 ‰) vs. the theoretically predicted values (alpha1=1.0026 and alpha2=0.9933).

Minor comments:
Line 65 – remove the word "be"

Revised as suggested.

Line 74 – approximations is misspelled

Revised as suggested.

Line 81 – notable should be notably

Revised as suggested.

Line 124 – ranged should be ranging

Revised as suggested.

Line 147 – remove the word "break" before NO2

Revised as suggested.

Line 419 – why mention nitrate aerosols specifically? Why would you not also compare with wet-deposited nitrate?

Revised as suggested.

Line 423-426 – need to be careful here. This addition was in response to the reviewers' who raised concerns about this. All of the calculations done in Figure 2 are based only on changes in NO and NO2 concentrations and ratios, using an alpha value that is constant. So it is important to consider that it is possible that temperature could have an impact on the interpretation of field data, and this should be investigated based upon comparisons between field collections and the laboratory experimental determinations.

Revised as suggested.

Figure 3 caption – you might add a note that points out the different y-axis for figure 3D.

Revised as suggested.

**List of changes:**

1. We changed Fig. 1 and its caption. We put the sketch as Fig. 1A and revised the captions.

2. We reported the RMSE between the LCIE line and our measured values. We suggest that the LCIE line with an LCIE factor of -10‰ gives the lowest RMSE (line 261).

3. We pointed out our study can be applied not only to nitrate aerosols but also dissolved nitrate in wet deposition (line 420).

4. We pointed out that the temperature may play an important role controlling the isotopic fractionations in the ambient environment (lines 424-428)

5. Other typo and grammar errors.

[revised manuscript text omitted]

---

## Author Response (AR3)

**Response to editor comments**

Dear Dr. Kaiser,

Thank you for your valuable input to this manuscript. Your suggestions significantly enhanced the overall quality of our paper. Please see below for a detailed point-by-point response.

1) Please integrate the supplement into the main text as appendix or appendices. The ACP manuscript guidelines require that "Supplementary material is reserved for items that cannot reasonably be included in the main text or as appendices. These may include short videos, very large images, maps, CIF files, as well as short computer codes such as matlab or python script. [...] Normal size figures, tables, as well as technical or theoretical developments that do not need to be included in the main text should be included as appendices."

Thanks for pointing this out. We moved the supplementary material into appendices and main text, now as Appendix A, B and C. We also numbered the figures and equations according to the format guideline.

2) Please remove the unnecessary approximations made in deriving equations 7 and 8, and make them consistent with Eq. 3. I have included the final steps of the calculation without approximation here (using the same $\delta$ and $\varepsilon$ symbols as in the calculation I sent you during the discussion stage). Obviously, your steps up to the ratio $R/R_2$ are fine (except that I have used the inverse ratio, to simplify the final step of the calculation): Setting $A = 0$ (very fast isotope exchange) immediately reproduces Eq. 3 for verification.

Thanks for this advice and we agree removing the approximations will making Eq. 2 & 3 consistent with Eq. 7 &8. In Eq. 7 & 8, we get rid of the approximation that assuming $\delta(NO_2)$-$\delta(NO)$=$R(NO_2)/R(NO)$. The non-approximated equation became:

$$\delta(NO_2)\text{-}\delta(NO)=\frac{(\alpha_2-\alpha_1)\times A+(\alpha(NO_2-NO)-1)}{\alpha_1 A+1}(1+\delta(NO))$$

Here, when A=0, Eq. 7 become Eq. 2, therefore the two sets of equations are consistent.

However, we suggest the approximation that $\alpha_1*A+1=A+1$ can still be applied since it introduces very little error and will greatly simplify the equation by reducing the number of unknowns. We calculated the $\delta(NO_2)$-$\delta(NO)$ values before and after this approximation by assuming $\alpha_1$=1.005, which is an overestimation because our theoretical calculations show that it should only range from 1.0025 to 1.0029. Even under this assumption, we can calculate that the differences between approximation vs. non-approximation was less than 0.05‰ (with A ranging from 0.001 to 500). This difference is much smaller than the analytical uncertainty in many labs so it can be neglected. In the meantime, if we use this assumption ($\alpha_1*A=A$), we can mathematically reduce the number of unknowns from two ($\alpha_2$-$\alpha_1$ and $\alpha_1$) to one (treating $\alpha_2$-$\alpha_1$ as one number). This approximation emphasized that the $\alpha_2$-$\alpha_1$ value, or the LCIE factor, acted as one important factor. Therefore, we suggest applying this approximation can simplify our model without bringing much uncertainty.

3) There is no need to repeat the reaction equations (R1 to R6, p. 5 of current supplement) or the definition of delta values (p. 6 of current supplement, lines from "Next, to calculate ..." to "$R_{NO2}/R_{NO} - 1$") in the Appendix.

In the appendix, we removed R1-R6 and referred the reactions to the main text. We also removed the delta definition equations.

4) Please remove unnecessary factors ("1000 ‰") from your mathematical equations (e.g. l. 180 and 182).

Revised as suggested.

5) Please also remove the unnecessary multiplication symbols ($\times$). These make the equations unnecessarily difficult to read. Such multiplication symbols are rarely required outside scientific notation, e.g. $8 \times 10^{-14}\,cm^3\,s^{-1}$.

Revised as suggested.

6) L. 182: The quantity with the symbol $\varepsilon$ should not be called "isotope enrichment factor" (which is a different kind of quantity). Please use the term "equilibrium isotopic fractionation" instead (as you do on l. 89 and elsewhere).

Thanks for pointing this out. In the text, we replaced all $\varepsilon$ symbols with $\alpha$-1 since we do not really need this extra symbol. Then, we name $\alpha$ "equilibrium fractionation factor"

7) L. 188 & 193: The quantities should be enclosed in parentheses so that the unit applies to value and uncertainty, e.g. "$(-58.7 \pm 0.8)$ ‰"

Revised as suggested.

**List of changes**

1. The supplementary material now is merged into the main text or as appendices.

2. A more precise equation 7 is used in the main text.

3. Improvement was made in mathematical expressions, such as removing "1000‰", "×", and including uncertainties in brackets.

4. Replaced symbol "$\varepsilon$" with "$\alpha$" to represent the equilibrium fractionation factor.

5. Reference list was updated to match the formatting requirement.

[revised manuscript text omitted]

---

## Author Response (AR4)

**Response to editor comments**

Dear Dr. Kaiser,

Thank you for your suggestions. I think I misunderstood some of your previous comments so there are some unclear expressions. Thank you for pointing them out, we have fixed them, which definitely improved the quality of our calculations as well as this manuscript.

For the supplementary information file, I did not upload anything in the last round. However, it seems that if we do not upload new file to overwrite the current supplementary file, it would remain unchanged and we cannot delete this file. Thus, in this draft I will upload a blank page to overwrite the current file, and if anything goes wrong, I will contact the editorial office to clarifying this problem.

In the meantime, I would like to point out that we spotted a calculation error in the previous manuscript. When calculating the EIE factor using the equations without approximation, I made a mistake when calculating $(1+\delta(NO_2))$. The $1+\delta(NO_2)$ values in our experiments should range from 0.948 to 0.956 but I accidentally made it 0.9948 to 0.9956. After fixing this mistake, we see the linear trend became more robust ($R^2$ increased from 0.96 to 0.9975) and the slope changed from 26.8‰ to 28.1‰. As a result, the EIE value became 28.9‰ instead of 27.5‰. We have revised this number throughout the manuscript. However, this change of 1.4‰ did not affect our entire story or the following LCIE calculations.

We re-calculated the LCIE using the equations without approximations after we changed the EIE value. The calculated LCIE value remain the same. Additionally, in order to show the consistency of the equations and the links between Fig. 1B and C, we revised Fig. 1B, C to better compare the two sets of data.

Additionally, I have added a table (Table 1) in the text to display all the data. However, I would like to mention that I deposited this data to the "open science framework" following the requirement on ACP website before we submitted the manuscript last year (please refer to the 'data availability' section in the end of the manuscript). The deposited data is almost the same as table 1, I hope it is OK.

**Please see below for a point-by point response to your comments:**

I noticed that you have now re-cast Eq. 8 as a function of three delta values, $\delta(NO)$, $\delta(NO_2)$ and $\delta(NO_x)$. This does not make sense because in your calculations in the Appendix, you are trying to eliminate one of the three unknowns using mass balance. Since you measure $\delta(NO_2)$ and assume that $\delta(NO_x)$ is constant, it would seem to make sense to cast the equations 8 (or 7) in terms of $\delta(NO_2)$ and $\delta(NO_x)$ – as you did in fact in the previous version of the manuscript. This also helps makes the link back to equation 3 and the link between Figs. 1B and 1C. I would therefore suggest you keep the previous version (using just the measured values, i.e. $\delta(NO_2)$ and $\delta(NO_x)$), with the corrections I pointed out before and make the axes of Figs. 1B and 1C consistent with equations 3 and a version of Eq. 8 using just these two variables, i.e.

$$\delta_2 - \delta_x = (1 + \delta_2)[1 - f(NO_2)]\frac{(\alpha_2 - \alpha_1)A + \varepsilon}{1 + A\alpha_2 + \varepsilon}$$

or (better)

$$\frac{\delta_2 - \delta}{1 + \delta_2} = \frac{1}{1 - f(NO_2)}\frac{\delta_2 - \delta_x}{1 + \delta_2} = \frac{(\alpha_2 - \alpha_1)A + \varepsilon}{1 + A\alpha_2 + \varepsilon}$$

This last version might be the most convenient because it explicitly shows the link between the y-axis label of Fig. 1C (slight recast from the version currently in the paper) and the conversion relation between the measured and the plotted values.

I am still not convinced that the approximation you are making ($1 + A\alpha_1 = 1 + A$) is necessary, convenient, or otherwise helpful. Actually, for large $A$ (which prevail at Summit, l. 409), the approximation you are making changes the asymptotic behavior of Eqs. 7 (and 8), which should be correctly

$$\lim_{A \to \infty}\frac{\delta_2 - \delta}{1 + \delta_2} = \frac{\alpha_2 - \alpha_1}{\alpha_2}$$

rather than

$$\lim_{A \to \infty}\frac{\delta_2 - \delta}{1 + \delta_2} = \alpha_2 - \alpha_1$$

In any case, the approximation is not necessary because you have to make assumptions about the magnitude of $\alpha_2$. Of course, you are right that the correction is small, but this is often the case for $\delta$ values and these small non-linearities are just due to the way $\delta$ values (relative isotope ratio differences) are defined. The approximations might have had their place when people were still doing calculations by hand (or maybe pocket calculator), but they don't really have any justification when everyone uses a computer. It would also simplify the paper if you removed the approximation (fewer equations and verbal explanation required). However, if you wanted to retain it, please give the correct solution and make it explicit what difference it makes when the approximation is applied, e.g. by writing Eqs. 7/8 as follows (using the correct approximation symbol $\approx$, not $\sim$):

$$\frac{\delta_2 - \delta}{1 + \delta_2} = \frac{1}{1 - f(NO_2)}\frac{\delta_2 - \delta_x}{1 + \delta_2} = \frac{(\alpha_2 - \alpha_1)A + \varepsilon}{1 + A\alpha_2 + \varepsilon} \approx \frac{(\alpha_2 - \alpha_1)A + \varepsilon}{1 + A}$$

We agree that we should use $\delta(NO_2)$, $\delta(NO_x)$ instead of $\delta(NO)$ in the figure, because these are the values we measured. We have revised the equations according to the calculation above and now we only used the measured values to plot Figure 1B and C. In the meantime, we provided both accurate calculation as well as approximation in Eq. 7 and 8. Figures 1 and 2 are also revised **using the equations without approximations**.

l. 136 & 155: Please use the correct approximation symbol $\approx$ (double wavy lines; not ~, which has other mathematical meanings).

Revised as suggested.

l. 113, 114 & 139: Please replace "ppb" with the SI unit "nmol mol$^{-1}$".

Revised as suggested.

l. 193: You mention that the method description includes three experiments that measured $\delta(NO_x)$, but the methods only refer to measurements of $\delta(NO_2)$. Are these the experiments when you titrate $O_3$ with NO? Please clarify this.

Thank you for pointing this out. In the previous version of the manuscript, we mentioned the measurement of $\delta(NO_x)$ after we discussed the setup of two sets of experiments. To clarify this question, we moved this description up and made it a new paragraph (lines 121-126 in the manuscript).

Finally, I noticed that a table is missing showing the experimental conditions and explicit results, which is required as per the journal's data policy. Please add such a table with a list of the 13 (?) experiments including initial and final NO, $NO_2$ and $O_3$ mole fractions, calculated $f(NO_2)$ values as well as the measured $\delta(NO_2)$ values.

We added a table (Table 1) in the manuscript after the figures. In the discussion, we referred to the table when discussing the results of our experiments.

**List of changes**

1. Equations 7 and 8 were revised to show the non-approximated equations as well as an approximated expression. The figures are also re-plotted to show the non-approximated results.

2. We fixed a previous miscalculation of the EIE value, which changed the EIE value from 1.0275 to 1.0289.

3. A table showing all the results is added.

4. Symbol '~' was replaced by '≈', unit 'ppb' was replaced by 'nmol mol$^{-1}$'.

[revised manuscript text omitted]

→ → Eq.

---

## Author Response (AR5)

**Response to reviewer**

Dear Dr. Kaiser,

Thank you for reviewing the revised manuscript and checking the deposited data. We have changed the ppb into nmol mol$^{-1}$ in the data table. Sorry for this mistake and thank you for pointing it out. Also, during last revision, we also revised the file on Open Science Framework (on June 24, same time as we upload the last draft) to make it as the same format as Table 1, so it might take a while for the system to process the file.

Additionally, we would like to thank Ms. Anna Wenzel, editorial support at ACP, for kindly helping us removing the latest supplementary material during the last revision. However, it seems that the old versions of the supplementary material were still in the systems. Therefore, in this round you might again see a supplementary file (which was an older version). Since we are not able to delete it from our end, please ignore the file, and if it causes further problem or confusion, we will contact the editorial office to remove it again.

**Here is our response to your comment about the EIE calculation:**

For the regression of our EIE calculation, we agree that an unforced linear regression for the five dark experiments will yield a slope of 33.9‰ instead of our proposed 28.1‰. It would also be nice if our results can show a good consensus with previous studies, especially from our own lab. However, we think it is probably not the case here.

This unforced regression yields an intercept of -2.5‰, which do not agree with the mathematical equation (Eq. 3). Theoretically, when $f(NO_2)=1$, the $\delta^{15}N$ value of $NO_2$ should be equal to $NO_x$ (because all $NO_x$ is in the form of $NO_2$), therefore the $(\delta(NO_2)-\delta(NO_x))/(1+\delta(NO_2))$ value should be 0. Referee 2 suggested that this -2.5‰ intercept might represent a systematic error ($^{17}O$ interference) when measuring the $\delta^{15}N$ values on the IRMS: when there is oxygen isotope mass independent fractionation, measuring the $\delta^{15}N$ will have a systematic error (approximately -1.5‰ when $^{17}O$ excess is 30‰). This error will become a problem if one used $NO_x$ gas with known $\delta(NO_x)$ value, then measure the $\delta(NO_2)$, in this case the intercept of the graph would not be 0. However, in our calculations, the $\delta(NO_x)$ values were also measured by the same method as $\delta(NO_2)$, it should have the same systematic error. As a result, **subtracting $\delta(NO_2)$ with $\delta(NO_x)$ should cancel out this systematic error**. Therefore, the intercept of this linear regression should be 0.

The above argument can be validated if we incorporate the data from "determining $\delta(NO_x)$" experiments into the linear regression. The conditions of these experiments were identical to the dark experiments except for the fact that we injected enough $O_3$ to convert 100% of NO into $NO_2$. Thus, they can be considered as three extra dark experiments. These three experiments presented three data points at $f(NO_2)=1$ ($1-f(NO_2)=0$) and $\delta(NO_2)-\delta(NO_x)=0\pm0.8$‰. **If we add these 3 points ((0, 0.8‰), (0, -0.1‰), (0, -0.7‰)) into Figure 2B, the unforced regression of the 8 points shows a slope of 28.5‰ with an intercept of -0.16‰. This slope is within the uncertainty of our forced linear regression slope.** The intercept here is much closer to 0 and should be originated from analytical uncertainties. Thus, we think the slope of the forced regression (28.1‰) should represent the true slope of this set of experiments.

We hope the above explanation clarified our reasons to calculate the slope by forcing the line through 0.

**List of changes**

1. in table 1, unit of NO, NO$_2$ and O$_3$ concentrations was changed from "ppb" to "nmol mol$^{-1}$"

[revised manuscript text omitted]

NO₂ (Δ(NO₂-NOₓ), **E-H**) at various $j$(NO₂), NOₓ level and $f$(NO₂) using Eq. (7) and (8). Each
panel represents a fixed $j$(NO₂) value (showing on the upper right side of each panel), and the
fractionation values are shown by color. Lines are contours with the same fractionation values, at
an interval of 5‰, the contour line representing 0‰ was marked on each panel except for A and
E.

[Figure]

**Fig. 3** $NO_x$ concentrations and calculated $\Delta(NO_2\text{-}NO_x)$ values at four sites. Stacked bars show the
NO and $NO_2$ concentrations extracted from monitoring sites (A-C) or calculated using 0-D box
model (D); the red lines are $\Delta(NO_2\text{-}NO_x)$ values at each site. Note that the $NO_x$ concentration (left-
y) axis on panel D is different from the rest.

| Experiment | Number | NO conc. (nmol mol$^{-1}$) | NO$_2$ conc. (nmol mol$^{-1}$) | O$_3$ conc. (nmol mol$^{-1}$) | $\delta$(NO$_2$) (‰) | f(NO$_2$) |
|---|---|---|---|---|---|---|
| Determining $\delta$(NO$_x$) | 1 | 0.0 | 17.8 | 13.4 | -59.5 | 1.00 |
| | 2 | 0.0 | 61.3 | 0.5 | -58.9 | 1.00 |
| | 3 | 0.0 | 18.9 | 10.7 | -58.0 | 1.00 |
| Dark experiments | 1 | 16.0 | 36.8 | 0.0 | -51.8 | 0.70 |
| | 2 | 33.6 | 28.8 | 0.0 | -43.9 | 0.46 |
| | 3 | 6.7 | 12.6 | 0.0 | -49.6 | 0.65 |
| | 4 | 16.2 | 16.9 | 0.0 | -45.1 | 0.51 |
| | 5 | 20.4 | 24.2 | 0.0 | -46.8 | 0.54 |
| Irradiation experiments | 1 | 7.1 | 6.4 | 2.8 | -47.5 | 0.47 |
| | 2 | 4.5 | 5.3 | 4.5 | -48.7 | 0.54 |
| | 3 | 3.3 | 4.4 | 4.2 | -49.8 | 0.57 |
| | 4 | 2.5 | 8.5 | 10.7 | -54.6 | 0.77 |
| | 5 | 5.2 | 18.1 | 11.0 | -54.0 | 0.78 |

Table 1. Experimental conditions, concentrations of NO, NO$_2$ and O$_3$ at steady state, and measured $\delta$(NO$_2$) values.

 **Appendix A. Chamber descriptions**

The chamber is a 10 m$^3$ Teflon bag equipped with several standard instruments including temperature and humidity probe, $NO_x$ monitor and $O_3$ monitor. 128 wall-mounted blacklight tubes surrounded the chamber to mimic tropospheric photochemistry and the photolysis rate of $NO_2$

(j($NO_2$)) when all lights are on have been previously determined to be $1.4\times10^{-3}$ s$^{-1}$, similar to a j($NO_2$) coefficient at an 81-degree solar zenith angle. The irradiation spectrum of the blacklights are shown in Figure A1. The chamber was kept at room temperature and one atmospheric pressure.

Before each experiment, the chamber was flushed with zero air at 40 L min$^{-1}$ for at least 12 hours to ensure the background $NO_x$, $O_3$ and other trace gases were below detection limit.

[Figure]

Figure A1 Spectral actinic flux versus wavelengths of the UV light source used in our experiments.

**Appendix B. Box model assessing the time needed for NO-NO$_2$ to reach isotopic equilibrium**

The time needed to reach NO-NO$_2$ isotopic equilibrium during light-off experiments were assessed using a 0-D box model. This box model contains only two reactions:

$^{15}NO_2 + ^{14}NO \rightarrow ^{15}NO + ^{14}NO_2$  k=8.14000 × 10$^{-14}$ cm$^3$ s$^{-1}$

$^{15}NO + ^{14}NO_2 \rightarrow ^{15}NO_2 + ^{14}NO$  k'=8.37525 × 10$^{-14}$ cm$^3$ s$^{-1}$

Where k and k' are rate constants of the reactions. The differences in rate constants were calculated by assuming an α(NO$_2$-NO) value of 1.0289. Six simulations were conducted at various initial NO (with δ$^{15}$N=0‰) and O$_3$ levels that were similar to our experiment. Then the δ$^{15}$N values of NO and NO$_2$ during the simulation were calculated from the model and were shown in Figure B1, suggesting that in our experimental condition, all systems should reach isotopic equilibrium within 1 hr.

[Figure]

Figure B1 Simulated NO-NO$_2$ isotopic equilibrium process in the chamber at various NO and O$_3$

concentrations.

**Appendix C. Deriving Equations 7 and 8**

When the system (R1-R6) reaches steady-state, we have:

$$d[^{15}NO_2]/dt = 0 \qquad\qquad \text{Eq. (C1)}$$

Therefore, using R1-R6:

$$k_1[^{15}NO_2][^{14}NO] + j(NO_2)\alpha_1[^{15}NO_2] =$$

$$k_5\alpha_2[^{15}NO][O_3] + k_1\alpha(NO_2\text{-}NO)[^{15}NO][^{14}NO_2] \qquad\qquad \text{Eq. (C2)}$$

From here we refer $^{14}NO_2$ and $^{14}NO$ as $NO_2$ and NO for convenience, rearrange the above equation, we get:

$$\frac{[^{15}NO_2]}{[^{15}NO]} = \frac{k_5\alpha_2[O_3] + k_1\alpha(NO_2-NO)[NO_2]}{j_{NO2}\alpha_1 + k_1[NO]} \qquad\qquad \text{Eq. (C3)}$$

Meantime, since the Leighton cycle reaction still holds for the majority isotopes (NO and $NO_2$), we have:

$$j_{NO2}[NO_2] = k_5[NO][O_3] \qquad\qquad \text{Eq. (C4)}$$

Thus,

$$\frac{[NO_2]}{[NO]} = \frac{k_5 \times [O_3]}{j_{NO2}} \qquad\qquad \text{Eq. (C5)}$$

From the text, when $j_{NO2} > 0$, we defined $A = \tau_{exchange}/\tau_{photo} = j_{NO2}/(k_1 \times [NO])$. Using the above equations, we know:

$$\frac{j_{NO2}}{[NO]} = \frac{k_5[O_3]}{[NO_2]} = Ak_1 \qquad\qquad \text{Eq. (C6)}$$

$$\frac{j_{NO2}}{k_1[NO]} = \frac{k_5[O_3]}{k_1[NO_2]} = A \qquad\qquad \text{Eq. (C7)}$$

Next, to calculate $\delta(NO_2)$-$\delta(NO)$, we use the definition of delta notation:

$$\delta(NO_2)\text{-}\delta(NO) = R_{NO2}/R_{std} - R_{NO}/R_{std} = (R_{NO2}/R_{NO}-1)(1+\delta(NO)) \qquad\qquad \text{Eq. (C8)}$$

$$\frac{R_{NO2}}{R_{NO}} = \frac{[^{15}NO_2][NO]}{[^{15}NO][NO_2]} = \frac{k_5\alpha_2[O_3][NO]+k_1\alpha(NO_2-NO)[NO_2][NO]}{j_{NO2}\alpha_1[NO_2]+k_1[NO][NO_2]}$$ Eq. (C9)

Divide both side by $k_1[NO][NO_2]$:

$$\frac{R_{NO2}}{R_{NO}} = \frac{\frac{k_5\alpha_2[O_3]}{k_1[NO_2]}+\alpha(NO_2-NO)}{\frac{j_{NO2}\alpha_1}{k_1[NO]}+1}$$ Eq. (C10)

Rearrange and substitute $\frac{k_5[O_3]}{k_1[NO_2]}$ and $\frac{j_{NO2}}{k_1[NO]}$ with A:

$$\frac{R_{NO2}}{R_{NO}} = \frac{\alpha_2 A+\alpha(NO_2-NO)}{\alpha_1 A+1}$$ Eq. (C11)

$$\frac{R_{NO}}{R_{NO2}} = \frac{\alpha_1 A+1}{\alpha_2 A+\alpha(NO_2-NO)}$$ Eq. (C12)

$$\frac{R_{NO}}{R_{NO2}} - 1 = \frac{(\alpha_1-\alpha_2)A-(\alpha(NO_2-NO)-1)}{\alpha_1 A+\alpha(NO_2-NO)}$$ Eq. (C13)

Thus,

$$\delta(NO_2)\text{-}\delta(NO)=\frac{(\alpha_2-\alpha_1)A+(\alpha(NO_2-NO)-1)}{\alpha_1 A+\alpha(NO_2-NO)}(1+\delta(NO_2))$$ Eq. (C14)

Then, using mass balance:

$$\delta(NO_2)f(NO_2)+\delta(NO)(1\text{-}f(NO_2)) = \delta(NO_x)$$ Eq. (C15)

We can derive Eq. 8:

$$\delta(NO_2)\text{-}\delta(NO_x)=\frac{(\alpha_2-\alpha_1)\times A+\alpha(NO_2-NO)-1)}{\alpha_1 A+\alpha(NO_2-NO)}(1+\delta(NO_2))(1\text{-}f(NO_2))$$ Eq. (C16)